# Nanosecond-resolution photothermal dynamic imaging via MHZ digitization and match filtering

Jiaze Yin [1,2,5], Lu Lan[1,2,5], Yi Zhang[3], Hongli Ni[1,2], Yuying Tan[4], Meng Zhang[1,2], Yeran Bai [1,2] & Ji-Xin Cheng [1,2,3,4✉]

Photothermal microscopy has enabled highly sensitive label-free imaging of absorbers, from metallic nanoparticles to chemical bonds. Photothermal signals are conventionally detected via modulation of excitation beam and demodulation of probe beam using lock-in amplifier. While convenient, the wealth of thermal dynamics is not revealed. Here, we present a lock-in free, mid-infrared photothermal dynamic imaging (PDI) system by MHz digitization and match filtering at harmonics of modulation frequency. Thermal-dynamic information is acquired at nanosecond resolution within single pulse excitation. Our method not only increases the imaging speed by two orders of magnitude but also obtains four-fold enhancement of signal-to-noise ratio over lock-in counterpart, enabling high-throughput metabolism analysis at single-cell level. Moreover, by harnessing the thermal decay difference between water and biomolecules, water background is effectively separated in mid-infrared PDI of living cells. This ability to nondestructively probe chemically specific photothermal dynamics offers a valuable tool to characterize biological and material specimens.

[1] Department of Electrical and Computer Engineering, Boston University, Boston, MA 02215, USA. [2] Photonics Center, Boston University, Boston, MA 02215, USA. [3] Department of Physics, Boston University, Boston, MA 02215, USA. [4] Department of Biomedical Engineering, Boston University, Boston, MA 02215, USA. [5] These authors contributed equally: Jiaze Yin, Lu Lan. ✉email: jxcheng@bu.edu

Photothermal microscopy is a versatile analytical tool to gauge optical absorption with extremely high sensitivity[1]. Unlike conventional spectroscopic methods that measure light attenuation, photothermal detection acquires absorption information by probing the thermal effect using a second light beam outside the absorption band. Its high sensitivity majorly benefits from the much-reduced background by employing a modulated heating beam and heterodyne detection of a frequency-shifted probe beam with a lock-in amplifier[2,3]. Using such a detection scheme, shot-noise limited imaging of single gold nanoparticles of 1.4 nm diameter has been demonstrated[2], and the single-molecule detection limit has been reported[4]. Recently, by employing a mid-infrared (mid-IR) laser as the pump source and visible light as the probe, a label-free vibrational spectroscopic imaging modality termed mid-infrared photothermal (MIP) microscopy has been demonstrated[5]. It obtains the mid-IR absorption contrast from a transient thermal field that is tightly confined in the vicinity of the absorber. By probing such a field with focused visible light, a submicron spatial resolution as good as 300 nm is achieved[6]. This new imaging modality not only enriches the photothermal techniques with enormous molecular fingerprint information, but also overcomes the limitations in conventional mid-IR absorption microscopy and near-field IR approaches. MIP microscopy is emerging as a valuable tool for biological and material science[7]. It enables IR spectroscopic imaging at the submicron spatial resolution, empowering a broad spectrum of applications in both research and industry, including but not limited to non-contact material characterization[8–10], biomolecular mapping[11,12], protein dynamics and aggregation in neurons[13,14], bacterial response to antibiotics[15], single virus detection[16] and imaging of metabolism in living cells and other organisms[5,17,18]. Since the first demonstration of high-performance MIP imaging in 2016[5], it has been quickly commercialized into a product[19], and the field expands with innovations, including wide-field detection[20], optical phase detection[21–23], photoacoustic detection[18,24], and synergistic integration with Raman spectroscopy[11].

Yet, despite the success in the development and applications of photothermal microscopy[1], the wealth and valuable information about an object's thermal dynamics along with the transient photothermal process is rarely explored. In principle, lock-in-based photothermal heterodyne imaging (PHI) can reveal thermal diffusivity of the medium through phase detection in the lock-in amplifier. This method[25,26] has enabled various applications, including observing superconducting transition[27], tissue differentiation[26,28], and revealing membrane interfaces[29]. However, lock-in demodulation at the fundamental modulation frequency loses the temporal resolution and all the higher-order harmonics of the photothermal signal. Therefore, this method is only capable of quantitatively retrieving thermal information for a well-defined model under ideal impulse or sinusoidal excitation, and it has a limited dynamic range for depicting the decay difference[30]. Thus, it is hard to use PHI to interpret a complex mid-infrared photothermal signal that originates not only from the absorbers but also from the embedding medium with distinct decay lifetimes. In the temporal domain, time-gated approaches, including digital boxcar averager or optical gating with a short pulse laser, can resolve the dynamics by tuning the delay between the probe point and excitation laser pulse[20,21,31]. Nevertheless, acquiring a complete thermal-dynamic profile depicting the temperature rise and decay at nanosecond resolution requires tedious, repetitive measurements, making it slow and unsuitable for non-repetitive transient processes.

In this work, we demonstrate a mid-IR photothermal dynamic imaging (PDI) system with nanosecond-scale temporal resolution and covering a bandwidth larger than 25 MHz. By using a wideband voltage amplifier and a megahertz digitizer, the photothermal dynamic response to a single IR pulse excitation is recorded. Combined with digital signal processing to filter out the noise outside the fundamental IR modulation frequency and its harmonics, PDI achieves more than a four-fold improvement in signal-to-noise ratio (SNR) than lock-in-based PHI. With these improvements, mid-IR PDI allowed high throughput metabolism analysis at the single bacterium level with an imaging speed nearly two orders of magnitude faster than the lock-in counterpart. Moreover, mid-IR PDI retrieves the transient thermal field evolvement and gives out information on the target's physical properties and micro-environment. Using PDI, we mapped the photothermal dynamics of various organelles inside a cancer cell. Distinct from the macroscopic observation on the homogeneous thermal response of tissue or cell as a whole[26,31], we show a highly heterogeneous, chemical-dependent thermal environment inside a cell. Furthermore, by harnessing the thermal decay difference between water and biomolecules, cellular components that are buried by the water background in conventional PHI can be differentiated in PDI based on their time-resolved signatures. Collectively, PDI enables direct detection of the transient photothermal process with nanosecond-level temporal resolution. Together with mid-IR excitation, PDI as a new approach allows for nondestructive investigation of a sample's intrinsic chemical and physical properties and is broadly applicable to biology and material science.

## Results

**Modeling the transient mid-infrared photothermal effect.** The photothermal phenomenon originates from the transformation of absorbed photon energy into heat through a nonradiative relaxation[31]. Under a pulsed laser excitation with a duration shorter than the thermal relaxation time, the absorbed energy forms a localized thermal field around the absorber. This thermal field induces concurrent thermoelastic deformation that modifies the local optical refractive index through local density change, which can be detected as a time-dependent photothermal signal through optical scattering. Compared with PHI detection of nanoparticles[32], the MIP thermal dynamics are distinct in two aspects. First, the MIP absorbers, specifically in the living system, cannot be simply modeled as point heat sources in a medium. For instance, targets like lipid droplets, protein aggregation, cytoplasm are bulky. The thermal dynamics are affected by both the absorbers and local medium collectively. Second, given the water absorption in the mid-IR range, both the absorbers and the aqueous medium will experience temperature elevation, which affects the contrast.

In a thermo-conductive environment, the evolution of an absorber's local temperature ($T$) under a mid-IR pump is governed by the heat transfer equation[33]:

$$mC_s \frac{dT}{dt} = Q_{abs} - Q_{diss} \qquad (1)$$

where $m$ and $C_s$ represent the mass and specific heat capacity of the absorber; $dT/dt$ is the temperature change over time; ($Q_{abs} - Q_{diss}$) denotes the energy flux, representing the rate difference between the absorbed and dissipated energies. $Q_{abs}$ can be approximated by $I_{IR}(t)\sigma_{abs}$, where $I_{IR}(t)$ represents the incident IR intensity over the IR pulse; $\sigma_{abs}$ represents the IR absorption cross-section. The heat dissipation follows Newton's law, i.e., $Q_{diss}$ is driven by the temperature gradient and given by ($hS[T(t) - T_{env}]$), where $h$ and $S$ represent the heat transfer coefficient and effective transfer surface area from specimen to environment, respectively. ($T(t) - T_{env}$) is the time-dependent temperature difference between the absorber and ambient environment $T_{env}$. We note that the heat transfer model here is

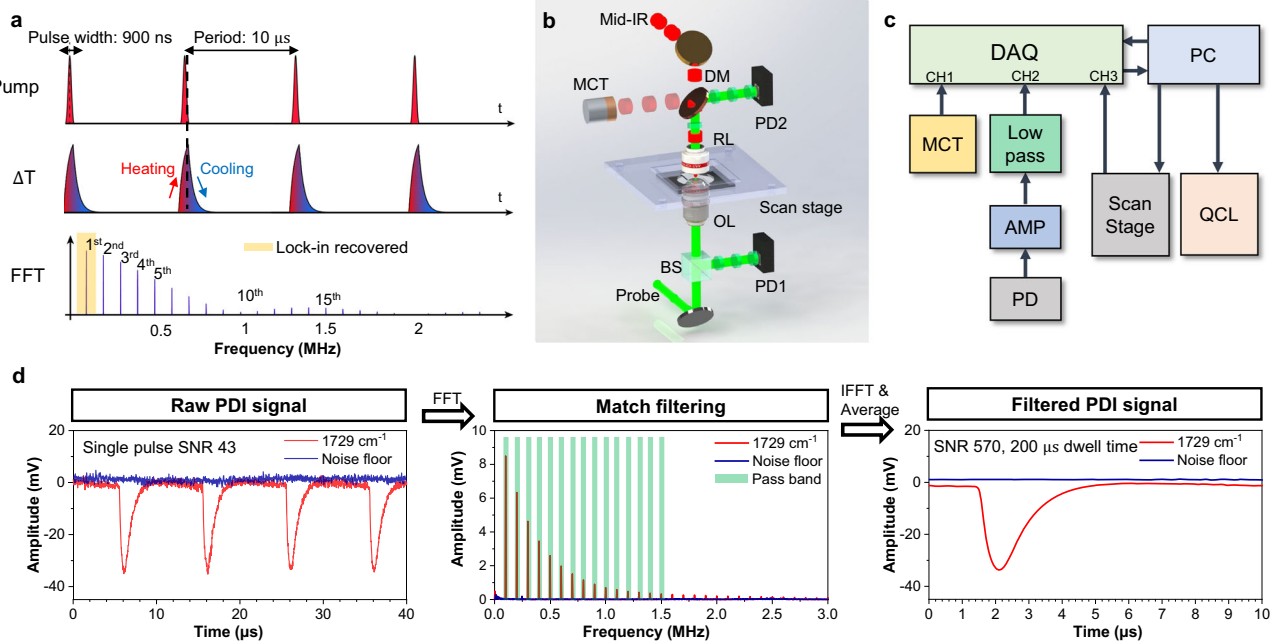

**Fig. 1 Principle and schematic of photothermal dynamic imaging. a** The simulated photothermal dynamics of a particle absorber under a pulsed IR pump. The absorber has a decay constant of 400 ns; the IR source has a repetition rate of 100 kHz and a pulse duration of 900 ns. **b** PDI setup schematics. A pulsed mid-IR beam is provided by a QCL and focused on the sample with a reflective objective lens (RL). The counter-propagated probe beam provided by a continuous-wave visible laser is focused by a water immersion objective lens (OL). Backscattered probe photons are collected with a 50/50 beam splitter (BS), forward scattered probe photons are collected by the reflective objective and separated by a dichroic mirror (DM). Both forward and backward probes are collected and sent to silicon photodiodes (PD) connected with a wideband voltage amplifier. A mercury cadmium telluride (MCT) detector is placed to monitor the IR pulse. **c** Electronics diagram. A high-speed data acquisition (DAQ) card is used to collect photodiode signal after voltage amplifier (AMP), MCT signal and sample position synchronously. A computer (PC) is used to control and synchronize different instruments. **d** Digital signal processing procedure. The sample was a PMMA particle with 500-nm diameter under 1729 cm⁻¹ excitation. Left panel, raw signal trace; middle panel, frequency domain distribution acquired via fast Fourier transform (FFT) and passband windows; right panel, filtered signal via inverse Fourier transform (IFFT). Match filtering is performed on the segment with 200 µs. Probe power on sample: 30 mW; IR average power measured before the objective lens: 5 mW.

valid when the time scale of the heat dissipation is longer than sub-nanosecond[34], which matches with the pulse duration of most mid-IR excitations.

During the heating process, $T(t)$ can be derived by solving Eq. (1) with the initial condition $T(0) = T_{env}$ and assuming constant IR irradiation $I_{IR}$ and ambient tempearture $T_{env}$:

$$T(t) = T_{env} + \frac{I_{IR}\sigma_{abs}}{hS}\left(1 - e^{-\frac{hS}{mC_s}t}\right) \quad (2)$$

When the IR pulse is finished, $Q_{abs}$ becomes zero. The temperature change is only driven by $Q_{diss}$. $T(t)$ is then solved as

$$T(t) = T_{env} + \left(T_{max} - T_{env}\right)e^{-\frac{hS}{mC_s}t} \quad (3)$$

Here, $T_{max}$ is the maximal temperature of the absorber after heating ends. From this model, both the heating and cooling process can be described as exponential processes. As an analogy to a resistor–capacitor (RC) circuit, $mC_s$ can be considered as the thermal capacitor and $1/hS$ as the thermal resistor[33]. The interplay of these two items results in a time constant $\tau = mC_s/hS$. A simulation of this transient process is illustrated in Fig. 1a middle row, assuming $\tau = 400$ ns. For absorbers with large heat capacity, such as bulky water and large particles, their time constants are expected to be large. Meanwhile, $hS$ is mostly related to the heat transfer capability of the embedding medium and the geometry of absorbers. Therefore, the thermal response is tightly connected to the physical properties of both the absorber and the environment, which can vary greatly in a heterogeneous system like a biological cell. Note that a single exponential decay model is employed for illustration. For complex sample configurations, where heat conduction inside

absorber or micro-environment needs to be considered, the assumption that constant ambient temperature $T_{env}$ no longer holds. For those cases, the transient photothermal dynamics has a superimposed decay with multiple lifetimes, and models that study complex decays in other fields, such as fluorescent lifetime imaging, can be applied[35].

In the frequency domain, the photothermal signal produced by IR pulses at a repetition rate of $f_{IR}$ can be treated as a Fourier synthesis of Eqs. (2) and (3) and contains harmonics of $f_{IR}$, as shown in the simulation of absorbers with $\tau$ of 400 ns (Fig. 1a, bottom row). Harmonics are spread out widely in the frequency domain, whose amplitude and phase delay can be described with the frequency response function $H(f)$ by analogy with a RC circuit. The relative amplitude of harmonic at the frequency $f$ with given decay constant $\tau$ is calculated by:

$$|H(f)| = \left|\frac{1}{1 + j2\pi\tau f}\right| \quad (4)$$

Large $\tau$ results in weak harmonics at high frequency, whereas fast decay signals exhibit strong harmonic components. The conventional lock-in only recovers the amplitude of the first harmonic at low frequency and misses all higher harmonics, which not only sacrifices the sensitivity[36] for low duty signal detection, but also causes contrast distortion in a heterogeneous sample with different thermal responses. Latterly we show that in the mid-infrared PHI of living cells using lock-in, the water background maximizes at the first harmonic, and it overwhelms the weak signal from the small organelles. To address these difficulties, we have developed a PDI system that is able to detect

both fundamental and harmonic components in the frequency domain.

**PDI instrumentation and data processing**. A schematic of our PDI system is shown in Fig. 1b. A pulsed mid-IR pump beam is provided by a quantum cascade laser (QCL) and focused on the sample with a reflective objective lens (RL). A counter-propagated probe beam from a continuous-wave 532 nm laser is focused by a water immersion objective lens (OL). Backscattered probe photons are collected with the same OL and separated with a 50/50 beam splitter; forward scattered probe photons are collected by the reflective objective and separated by a dichroic mirror. Both forward and backward probe photons are collected and sent to silicon photodiodes (PD) connected with a wideband voltage amplifier worked in AC coupling (10 Hz–100 MHz). In the following experiments, the photothermal signal was detected on backward scattering for nanoparticle and bacteria samples; on forward scattering for cell samples. The voltage signal is filtered with a low pass filter with a cut-off frequency of 25 MHz and sent to a high-speed digitizer (DAQ) with a sampling rate of 50 million samples per second. Meanwhile, a mercury cadmium telluride (MCT) detector is placed to monitor the IR pulse, and the signal is digitized by the same DAQ synchronously. The electronics diagram is shown in Fig. 1c.

In PDI, we process the acquired data following the procedure shown in Fig. 1d. The PDI raw signal (left panel) was acquired from the center of the polymethyl methacrylate (PMMA) particle with a diameter of 500 nm under the IR pump at its absorption peak of 1729 cm$^{-1}$. Benefited from the broadband detection scheme, a single pulse photothermal signal can be clearly resolved with an SNR over 43 without averaging. During the imaging process, the signal acquired per pixel is a segment with multiple pulses. We further performed match filtering on each segment with a comb-like passband in the frequency domain. In the comb passband, each small window has its center position colocalized at the harmonic frequencies to reject most of the non-modulated noise (middle panel). After filtering and inverse Fourier transform, the photothermal dynamic signal was obtained in the temporal domain with a much-improved SNR of 570 (right panel). By scanning the sample, an X-Y-t stack with each spatial pixel extended in the temporal domain is obtained. The single pulse resolved capability in PDI enables an unprecedented imaging speed with pixel dwell time as short as 50 μs, practically limited by the stage scanning speed.

**Mid-infrared photothermal dynamic imaging of nanoparticles.** To characterize PDI's performance and evaluate its capability of scrutinizing transient thermal dynamics with chemical specificity, we performed PDI on PMMA particles with a nominal diameter of 300 nm. The QCL is set with a repetition rate of 100 kHz and a pulse duration of 600 ns. By tuning the QCL to 1729 cm$^{-1}$, corresponding to the absorption peak of the C=O bond in PMMA, we acquired the photothermal dynamics signal (Supplementary Movie 1). The IR pump starts at $t_1 = 0$ ns. As in Fig. 2a, most of the single particles with absorption reached their highest temperature at $t_2 = 440$ ns. The photothermal contrast disappears under an off-resonance IR pump at 1600 cm$^{-1}$ (Fig. 2b). To compare PDI with PHI, we acquired the intensity image under 1729 cm$^{-1}$ at the same field of view by demodulating at 100 kHz using the lock-in (Fig. 2c). The line profiles across the indicated particle in Fig. 2a, c are shown in Fig. 2d. For PDI, an SNR of 230 was achieved with a pixel dwell time of 200 μs, while it was 52 using PHI, corresponding to more than four-fold improvement in detection sensitivity. The SNR

improvement majorly comes from two aspects. Firstly, the laser noise level at high frequency is largely reduced as shown in Supplementary Fig. 1, and the transient photothermal signal has significant components at high frequency. Secondly, the harmonics of signal constructively add up, while the uncorrelated random noise does not. A quantitative analysis of SNR improvement is given in Supplementary Note 1. For the above experimental condition, a theoretical SNR gain of 5.4 is expected. Experimentally, our result shows an improvement of 4.3 times, which is close to theory. The small discrepancy is attributed to the non-ideal impulse shape of the actual IR excitation.

Apart from the sensitivity improvement, PDI concurrently revealed dynamics that enable accurate thermal property characterization. The photothermal dynamics of the nanoparticle indicated in Fig. 2a under 1729 cm$^{-1}$ excitation is presented as the red curve in Fig. 2e. The photothermal induced scattering modulation is known to be linear dependent on temperature[6,37]. The raw signal has a negative sign, indicating diminished backscattering as temperature rising. Here, an inversed waveform is used for better illustration. The temporal resolution in our system is ultimately limited by the photodiode response time, which is a few nanoseconds. With that high temporal resolution, the time-resolved energy flux function $[Q_{abs}(t) - Q_{diss}(t)]$ in Eq. (1) could be directly evaluated by taking the derivative of PDI signal to time, shown as the blue curve in Fig. 2e. The thermal-dynamic process is composed of three stages. At the beginning of heating (from $t_1$ to $t_2$), the heat dissipation was negligible. $Q_{abs}(t)$ related to energy absorption was dominant, and it resulted in a waveform similar to the IR pulse shape $I_{IR}(t)$ shown as the black curve in Fig. 2e. The temperature kept rising until the heat dissipation term equaled to heat influx; at this point, the net energy flux became zero, and the absorber entered a thermal equilibrium state. From $t_2$ to $t_3$, with the IR intensity reduced gradually, the dissipated energy became dominant, and the energy flux function became negative, showing that the absorber entered the cooling stage. After the IR pulse ($>t_3$), the heat flux function only had the heat dissipation term $Q_{diss}(t)$, shown as an exponential decay.

Subsequently, we measured the thermal decay constant of the pure dissipation process after $t_3$. Here we extracted the exponential decay constant with the least square fitting method. Note that advanced methods, including maximum likelihood estimation[38] and maximum entropy methods[39,40], can be exploited to obtain the decay information. The fitted exponential decay function is shown as the green dashed line in Fig. 2e, which has a decay constant of 300 ns. From Eq. (3), this time constant is given by $mC_s/hS$. With the material's density $\rho$ and $C_s$ from reference[41], we can determine the heat transfer parameter $hS$. For this 300 nm PMMA particle on a CaF$_2$ substrate in air, the heat transfer parameter is derived to be 7.78E−8 W/K. To verify the experimental result, this parameter is independently calculated to be 7.65E−8 W/K using the finite-element method (FEM) (Supplementary Fig. 3), which closely matches the experimental measurement. The thermal decay constants of PMMA particles with various sizes were statistically studied by PDI (Supplementary Note 2), as shown in Supplementary Movie 2 and Supplementary Fig. 4. The particles with diameters of 300 nm and 500 nm showed distinct thermal decay constants of 280 ns and 495 ns, respectively. In addition, by acquiring complete photothermal dynamics together with scattering intensity, PDI enables transient temperature rise measurement according to the modulation depth as presented in Supplementary Note 3. For $D = 500$ nm PMMA particle at its absorbing peak 1729 cm$^{-1}$, the highest transient temperature rising is estimated to be 7.6 K. Such temperature rise less than 10 Kelvin and with a duration less than

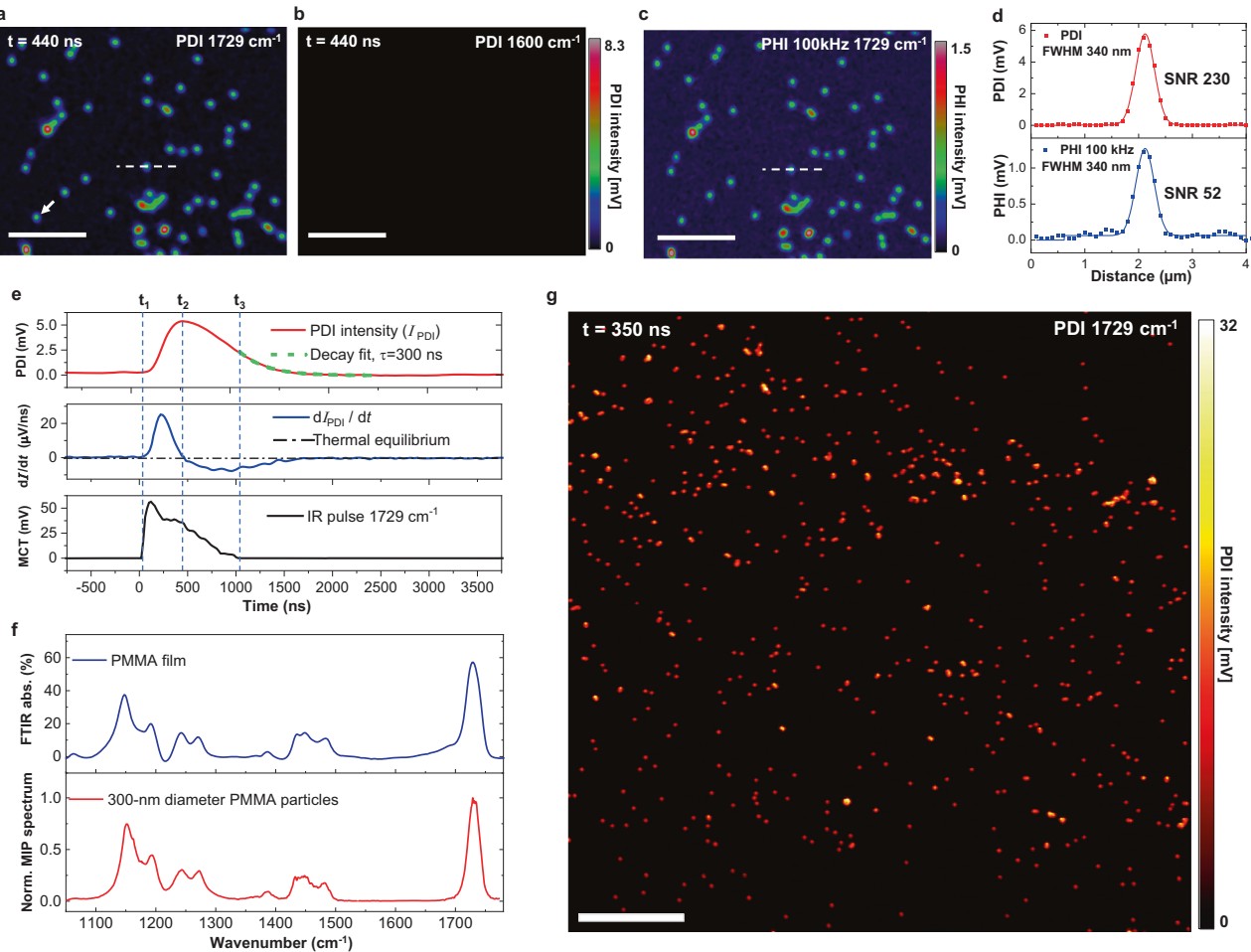

**Fig. 2 Mid-infrared photothermal dynamic imaging and spectroscopy of PMMA nanoparticles. a, b** PDI-acquired photothermal intensity image of 300-nm diameter PMMA particles at the absorption peak at 1729 cm$^{-1}$ and off-resonant wavelength at 1600 cm$^{-1}$, respectively. **c** MIP intensity image at 1729 cm$^{-1}$ acquired with the lock-in amplifier in the same field of view. **d** Line profiles across the particle indicated in (**a**) and (**c**). Pixel dwell time: 200 μs; Step size: 100 nm; Probe power on sample: 30 mW; IR power on sample: 5 mW at 1729 cm$^{-1}$, 16 mW at 1600 cm$^{-1}$; IR repetition rate: 100 kHz; IR pulse duration: 600 ns at 1729 cm$^{-1}$, 900 ns at 1600 cm$^{-1}$. Scale bars: 5 μm. **e** Photothermal dynamics of the particle indicated in (**a**). The red curve is the transient PDI intensity. The green dash line is the fitted exponential decay function, with a decay constant $\tau$ of 300 ns. The blue curve is the derivative of PDI intensity over time. The dash-dot line indicates a thermal equilibrium state. The black curve is the synchronously acquired IR pulse. **f** Spectrum fidelity. FTIR spectrum of a standard PMMA film (top) and normalized MIP spectrum profile measured at the indicated 300-nm diameter PMMA particle (bottom). Acquisition speed: 50 cm$^{-1}$ per second. **g** Large area PDI of 500-nm diameter PMMA particles at 1729 cm$^{-1}$. Pixel dwell time: 100 μs; Step size: 150 nm; Probe power on sample: 25 mW; IR average power measured before objective lens: 11 mW; IR repetition rate: 500 kHz; IR pulse duration: 400 ns. Scale bar: 20 μm.

hundreds of nanosecond is biologically safe. Next, we tuned the IR wavelength and recorded the photothermal spectrum of the indicated particle, as shown in Fig. 2f. When comparing it with the spectrum of PMMA film acquired with FT-IR, good consistency, such as the peak ratio, was observed in the entire fingerprint region. Collectively, with the capability of characterizing thermal decay properties with submicron spatial resolution and spectral fidelity, PDI offers a new dimension for molecular analysis.

Importantly, the retrieved decay constant of absorbers can be used for optimizing the IR modulation frequency and increasing the imaging speed accordingly. For 500 nm PMMA particles with a decay constant of 495 ns, more than 95% heat drain out at the initial 1500 ns of the cooling process. Thus, modulation frequency higher than 100 kHz used in the previous work[5] can be employed without influencing the photothermal signal amplitude. Figure 2g shows PDI on PMMA particles with a diameter of 500 nm at an IR repetition rate of 500 kHz and pulse duration of 400 ns. The pixel dwell time of PDI was reduced to 100 μs, and average single-

particle SNR over 261 was achieved. This imaging speed is 30 times faster than the previous report work[11] while achieving similar SNR. With such improvement, we further explored high-speed PDI of biological specimens, as shown below.

**PDI enables high-speed metabolic analysis at a single bacterium level.** Given the cell-to-cell heterogeneity, high-speed imaging is essential to understand cell functions and pathways[42]. For example, sensitive single bacterium metabolism analysis plays an important role in synthetic biology where rare species of interest need to be differentiated from a large population[43], or determination of antimicrobial susceptibility from few cells[44], such as cells from a small volume of cerebrospinal fluid. Here, we explored PDI imaging of carbohydrate conversion into biomass at a single bacterium level using *E. coli* as a testbed.

We first implemented the PDI of *E. coli* at 100 kHz IR modulation frequency, and 200 μs pixel dwell time. The thermal response of a single *E. coli* is shown in Fig. 3a. The decay constant

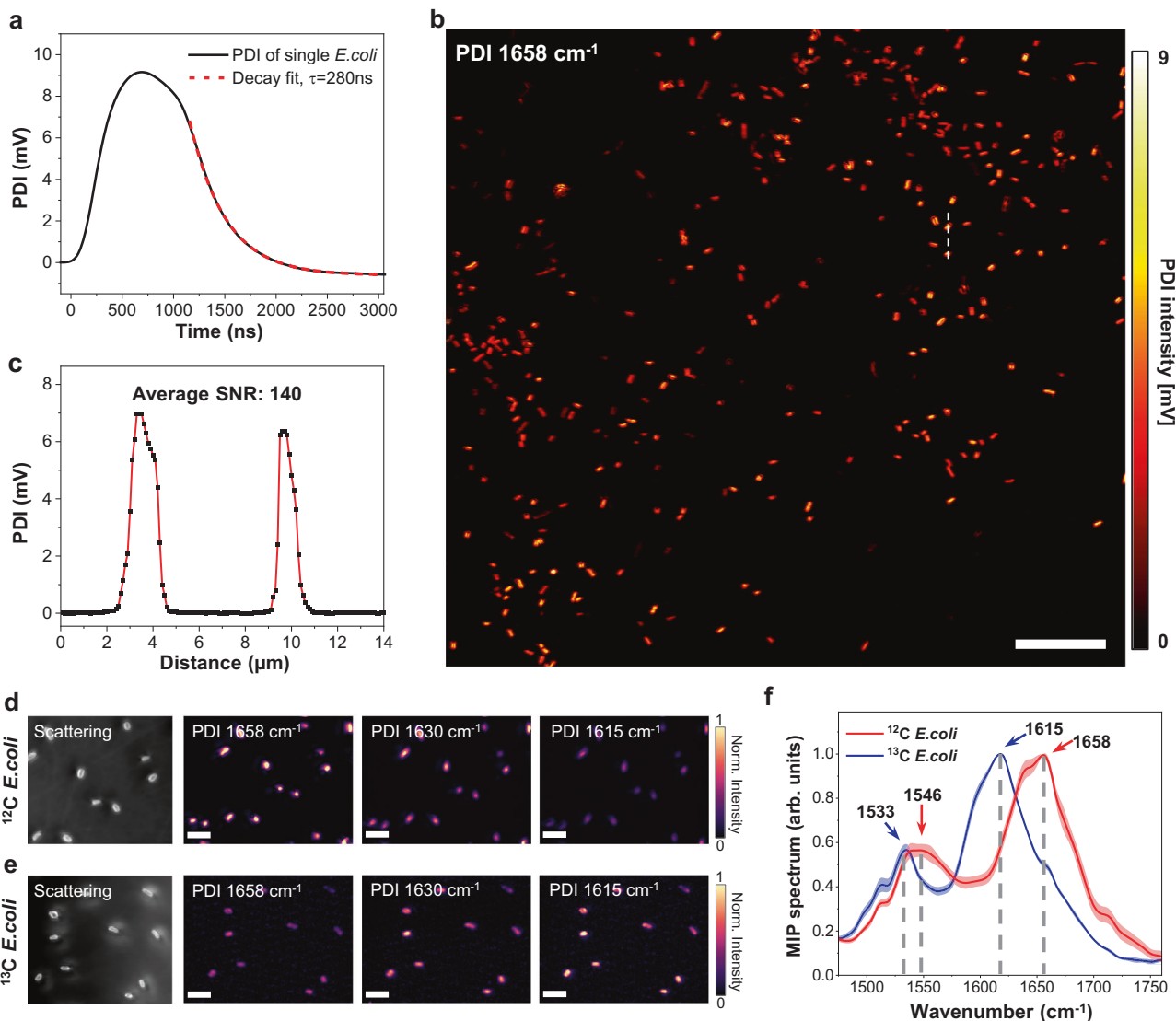

**Fig. 3 PDI enables high-throughput metabolic imaging at a single bacterium level. a** Photothermal dynamics of single *E. coli* under 1658 cm$^{-1}$ excitation. Measured decay constant $\tau$ of single bacteria is 280 ns. **b** High-speed PDI of *E.coli* at the protein Amide I band. Pixel dwell time: 50 μs; Scale bars: 20 μm. Probe power on sample: 10 mW; IR average power measured before the objective lens: 35 mW at 1658 cm$^{-1}$; IR repetition rate: 1.0 MHz; IR pulse duration: 300 ns. **c** Line profile of indicated line in (**b**). **d** PDI of $^{12}$C glucose-treated *E. coli* at Amide I band. (e) PDI of $^{13}$C glucose-treated *E. coli* at Amide I band. Amide I peak is shifted from 1658 cm$^{-1}$ to 1615 cm$^{-1}$ due to the isotope substitution. Scale bars: 5 μm. (f) MIP spectra acquired from $^{12}$C and $^{13}$C glucose-treated *E. coli* (Error bands represent standard deviation of the mean).

is found to be 280 ns, where 95% heat drains out within 800 ns. Such thermal property allows IR modulation at up to 1 MHz without overheating. At 1 MHz frequency and with a pixel dwell time of 50 μs, we performed PDI of *E. coli* under 1658 cm$^{-1}$ excitation, corresponding to the Amide I band, as shown in Fig. 3b. An SNR over 140 is achieved for a single cell (Fig. 3c). This speed is 600 times faster than the previous lock-in-based MIP imaging of single *E. coli*[6]. The PDI image covering more than 350 individual cells was acquired within 70 s, showing its high throughput.

Next, we implemented the PDI of *E. coli* treated with normal and isotope-labeled glucose. Under the $^{12}$C-glucose carbon source, the Amide I absorption peak is localized at 1658 cm$^{-1}$, as shown in Fig. 3d. By substituting the carbon source with $^{13}$C-glucose in the growth medium, the Amide I band is shifted to 1615 cm$^{-1}$, as shown in Fig. 3e. The PDI spectral profiles of these two different groups of *E. coli* are shown in Fig. 3f. It is clear that the Amide I peak is shifted from 1658 cm$^{-1}$ to 1615 cm$^{-1}$, and

the Amide II peak is shifted from 1546 cm$^{-1}$ to 1533 cm$^{-1}$. Collectively, these data demonstrate that PDI has high throughput and high sensitivity for single-cell metabolism imaging.

**PDI separates biomolecular signal from water background in infrared photothermal imaging of living cells.** Due to water absorption in the entire fingerprint region, the culture medium experiences temperature modulation when performing MIP imaging of living cells. Bulky water is known to have a large heat capacity, which helps increase the MIP contrast between biosamples and background[5,21] to some extent. However, in the conventional lock-in-based MIP system, using signal amplitude alone is not sufficient for separating weak biomolecular signals from water background. By capturing the photothermal dynamics, PDI can be used to separate the water contribution from the organelles' signal based on their distinct thermal responses.

To investigate the transient thermal responses of various organelles inside cells, we performed PDI of U87 brain cancer

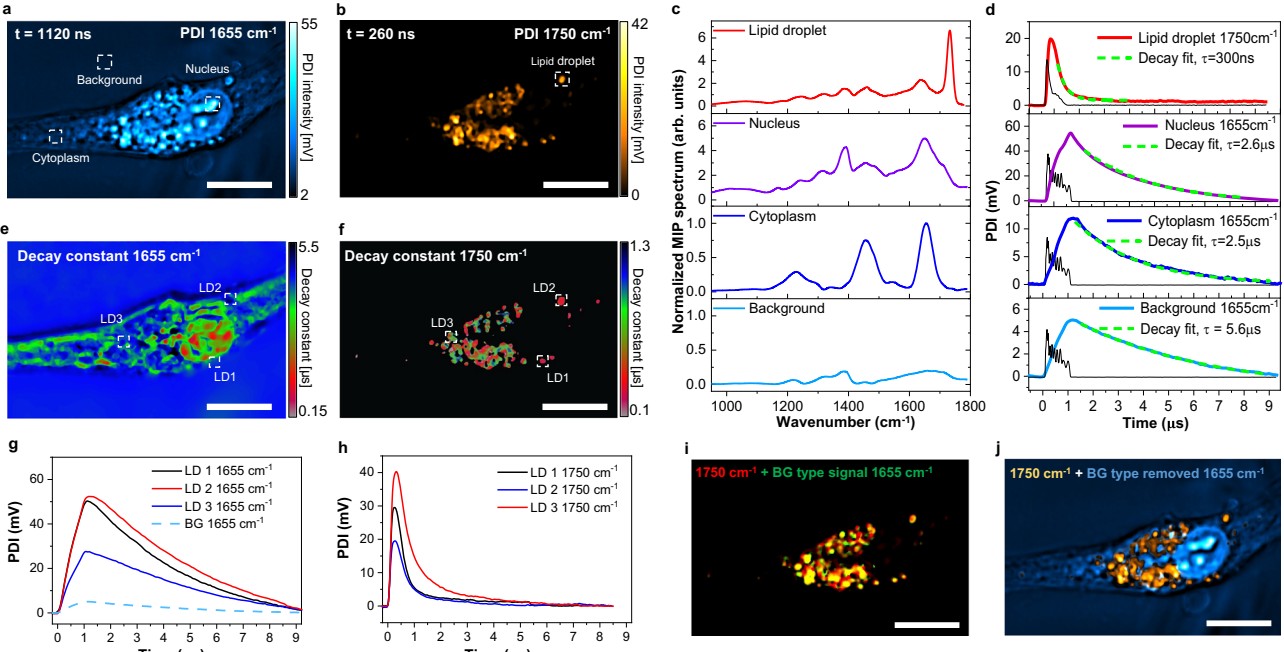

**Fig. 4 Mid-infrared photothermal dynamic imaging of U87 brain cancer cells. a** PDI of U87 brain cancer cells at 1655 cm$^{-1}$ Amide I band. **b** PDI of U87 brain cancer cell at 1750 cm$^{-1}$ lipid C=O band. **c** MIP spectra of locations indicated in (**a**, **b**). All spectrum intensities are normalized according to the peak of cytoplasm at 1655 cm$^{-1}$. Acquisition speed: 50 cm$^{-1}$ per second. **d** Thermal dynamics at the positions indicated in (**a**, **b**). Black curves are synchronously acquired IR pulse. Green dash lines indicate fitted exponential decay functions and corresponding time constant $\tau$. **e** Decay constant map at 1655 cm$^{-1}$. **f** Decay constant map at 1750 cm$^{-1}$. **g** Thermal dynamics of background (BG) and positions indicated in (**e**) under 1655 cm$^{-1}$ excitation. **h** Thermal dynamics of lipid droplets (LD) indicated in (**f**) under 1750 cm$^{-1}$ excitation. **i** Merged intensity image at 1750 cm$^{-1}$ (red) with background type signal at 1655 cm$^{-1}$ (green). The yellow color shows a highly overlapped spatial distribution. **j** Merged lipid contents at 1750 cm$^{-1}$ with protein contents at 1655 cm$^{-1}$ after removing the background type signal. Pixel dwell time: 200 μs; probe power on sample: 20 mW; IR average power measured before objective lens: 22 mW at 1655 cm$^{-1}$, 2 mW at 1750 cm$^{-1}$; Scale bars: 10 μm.

cells in D$_2$O PBS. The D$_2$O PBS was used to maintain cell morphology and reduce the water absorption of mid-IR around 1650 cm$^{-1}$. By tuning the IR to 1655 cm$^{-1}$ corresponding to the Amide I band, protein-rich contents inside the cell give strong contrast in the photothermal intensity map shown as Fig. 4a. Due to the residual water absorption at this wavenumber, the background was observed, but it was relatively weak compared to the cells' signal[5,20]. In this protein map, uniformly distributed protein contents in the cytoplasm and a strong signal from the nucleolus are observed. By tuning the IR to 1750 cm$^{-1}$ corresponding to the C=O band from lipids, individual lipid droplets show a strong signal in Fig. 4b. To verify the chemical contents, we performed photothermal spectroscopy in the fingerprint region at lipid droplet, nucleus, cytoplasm, and medium, respectively, shown in Fig.4c. The spectra at the nucleus and cytoplasm show a strong peak in the Amide I band at 1655 cm$^{-1}$ and a shifted Amide II band at 1450 cm$^{-1}$ due to the deuterium substitution of N-H bond[45]. The spectrum for lipid droplets shows a strong peak at 1750 cm$^{-1}$ contributed by the C=O bond in esterified lipids. Interestingly, a broad peak centered at 1650 cm$^{-1}$ showed up in the spectra of lipid as well. It matched the result of the intensity map at 1655 cm$^{-1}$ (Fig. 4a), where lipid droplets are visible as well. This abnormal contrast at the protein band was found in previously reported scattering-based photothermal imaging but without in-depth explanation[22,46].

Here, enabled by PDI, we investigated the origin of this abnormal signal by studying the transient photothermal dynamics of various subcellular components indicated in Fig. 4a, b. The results in Fig. 4d show distinct thermal responses among organelles. The lipid droplets inside cells show a relatively fast

decayed signal with a time constant of 300 ns. Nucleolus and cytoplasm with rich protein contents show slower thermal dynamics on the level of 2.5 μs. The water background at 1655 cm$^{-1}$ has the longest decay with a decay constant larger than 5 μs, largely because of the large water heat capacity. For a more intuitive illustration, we generated the decay constant map of 1655 cm$^{-1}$ (Fig. 4e) and 1750 cm$^{-1}$ (Fig. 4f). From the decay map, the background and cellular structures are clearly differentiated for their distinct thermal response. The cytoplasm and nucleus have decay constant on the level of 2.5 μs with 1655 cm$^{-1}$ excitation (Fig. 4e). Lipid droplets have decay constants ranging from 150 ns to 500 ns under 1750 cm$^{-1}$ excitation (Fig. 4f).

Importantly, those "lipids" seen in the intensity image at 1655 cm$^{-1}$ (Fig. 4a) did not show fast dynamics (Fig. 4e). Instead, they exhibit a decay constant similar to that of the background. Since the detected signal originates from the modulation of the scattering field, the signal intensity is proportional to $(n_s - n_m)$, where $n_s$ and $n_m$ are the refractive index of organelle and medium, respectively. In the presence of water absorption, $n_m(t)$ becomes time-dependent, and such change results in a pseudo-MIP signal without organelle's absorption. To better understand the origin of "lipids" seen under 1655 cm$^{-1}$ excitation, underlying thermal dynamics at indicated positions in Fig. 4e and Fig. 4f are plotted in Fig. 4g and Fig. 4h, respectively. At 1750 cm$^{-1}$, lipids' signal has a fast response and decays in a few hundred nanoseconds. However, the signal at 1655 cm$^{-1}$ from the same regions of interest shows a relatively slow process with a decay constant higher than 5 μs, similar to that of the water background, as shown in the dash lines in Fig. 4g. In summary, by analysis of photothermal dynamics, we conclude that the

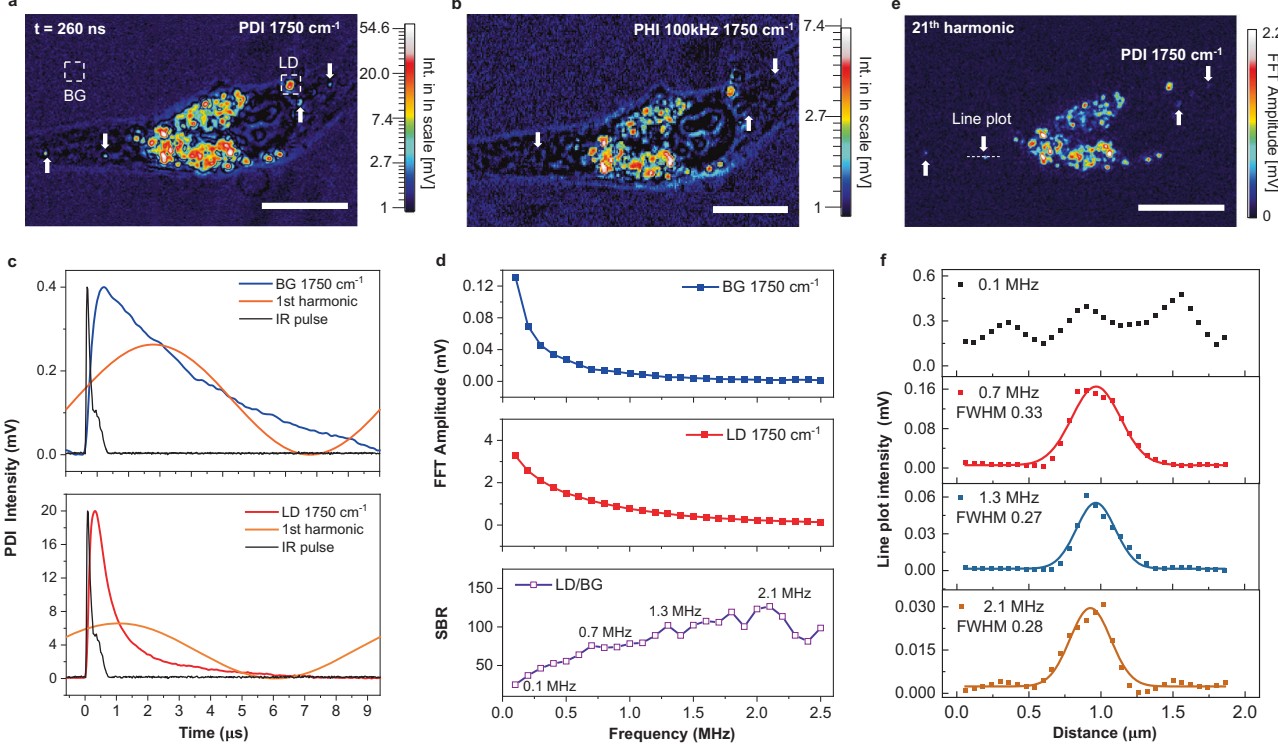

**Fig. 5 PDI detection of weak lipid signal from water background. a** PDI of U87 cancer cell at 1750 cm$^{-1}$, displayed with natural logarithm scale color bar. **b** MIP image of the same field of view at 1750 cm$^{-1}$ acquired by the lock-in method, displayed with natural logarithm scale color bar. Arrows indicated lipid droplets in (**a**) are hardly seen. **c** Photothermal dynamics of background (BG) and lipid droplet (LD) at the position indicated in (**a**). Black curves are synchronously acquired IR pulse. **d** Frequency domain representation of signals in (**c**) acquired by Fast Fourier transform (FFT) and corresponding signal to background ratio (SBR). **e** The 21st harmonic amplitude image was acquired by taking the Fourier transform of the PDI stack. **f** Intensity profiles of the line indicated in (**e**) at different frequencies. PDI pixel dwell time, 200 μs; PHI pixel dwell time 500 μs; Probe power on sample: 20 mW; IR average power measured before objective lens: 2 mW. Scale bars: 10 μm.

1655 cm$^{-1}$ peak in lipid (Fig. 4c) comes from the water background rather than the organelles.

Next, by utilizing the distinct thermal properties between water medium and lipid droplets, we successfully extracted those water-induced signals at lipid droplet positions with the acquired thermal dynamics via a simple algorithm that evaluates their photothermal intensity and decay constants, as shown in Fig. 4i. Those strong signals with background type thermal dynamics at 1655 cm$^{-1}$ are displayed in green color, and lipid droplets image acquired under 1750 cm$^{-1}$ excitation is shown in red color. The yellow contrast indicates highly matched colocalization in the merged image. After removing those water-induced pseudo signals at 1655 cm$^{-1}$ from the intensity image, a corrected content map between lipids and proteins is produced, as shown in Fig. 4j.

**PDI allows extraction of extremely weak signal from water background**. The above photothermal dynamic results show that lipid droplets exhibit a fast decay on the order of a few hundreds of nanoseconds, while the water background is much slower, on the order of a few microseconds. By capturing the high-order harmonic signals, PDI further enabled us to visualize the small lipids (Fig. 5a). These small lipids were completely buried in the water background when lock-in detection was used (Fig. 5b). To better understand this capability, we performed a Fourier analysis of the photothermal dynamic signals.

The transient photothermal signals of background and lipid droplets at indicated positions in Fig.5a are shown in Fig.5c. Their fundamental components (100 kHz) are acquired by Fourier transform and plotted as orange curves. Those signals of small

particles have fast responses and load high-order harmonic components. On the contrary, the water background is majorly localized at the fundamental modulation frequency. Thus, lock-in demodulation at the fundamental frequency minimizes the contrast between lipid droplets and the background.

The frequency responses of signals in Fig. 5c are shown in Fig. 5d. The slow background (blue square) majorly has components in the first and second harmonics. As a comparison, the fast lipid signal (red square) is widely spread out in the frequency domain, with the first harmonic only containing less than one-fifth of the total energy. Consequently, the lipid to background ratio (purple square) increases till the 21st harmonic. At the first harmonic that lock-in demodulates, the contrast is the lowest (Fig. 5d). Instead, the photothermal image (Fig. 5e) with the 21st harmonic (2.1 MHz) shows clear contrast for small lipids with minimal background. The intensity profiles of the line indicated in Fig. 5e at different frequencies are plotted in Fig. 5f. At the first harmonic (100 kHz), which is equivalent to lock-in extraction, the lipid signal is hardly resolved from the background signal, while a good signal to background ratio is shown in higher-order harmonics (0.7–2.1 MHz).

## Discussion

We have demonstrated a photothermal dynamic imaging microscope that can sense the transient photothermal modulation with nanosecond temporal resolution. This advanced technology enables concurrent detection of chemically specific IR absorption and physically specific thermal dynamics at submicron spatial resolution. We retrieved the thermal response of various organelles inside a cell for the first time. Our data shows that

cytoplasm, nucleus, and lipid droplets exhibit distinct time-resolved signatures. Based on the time-resolved signatures, PDI enabled the differentiation of small signals from water medium contribution.

Compared with conventional PHI via lock-in amplifier, PDI increases the sensitivity by more than four-fold for low duty cycle photothermal signals. This improvement leverages the broad detection bandwidth by capturing all the harmonics components induced by the short-pulsed IR pump in the high-frequency region where laser noise is reduced. Those signal harmonics are correlated and can add up, while the noise is uncorrelated and can be suppressed. For this reason, PDI can largely benefit the mid-IR photothermal microscope with a powerful optical parametric oscillator (OPO) source[20,47], which has a pulse duration of few nanoseconds and a fixed low repetition rate of tens kilohertz. Such a short-pulsed and high peak power excitation source is highly preferred for generating large modulation depth of small objects on a thermo-conductive substrate or in an aqueous environment, where heat dissipation is relatively rapid[7]. In such a case, the photothermal signal has a duty cycle of less than 1% and the lock-in amplifier can only capture a tiny portion of modulation or balanced detection for reducing the large laser noise is required at such low modulation frequency[47]. By capturing all the harmonics, PDI can improve the SNR over one order of magnitude as illustrated in Supplementary Note 1. Our results show that careful choice of detection method has to be made for optimal sensing of the photothermal signal induced by a short pulse excitation. While the same level of SNR improvement may be achieved by using other advanced demodulation techniques that are capable of capturing those high-frequency photothermal signals, such as signal synthesis using lock-in multi-channel demodulation[48,49] or boxcar averaging[50], those advanced demodulation methods are not yet widely available. In contrast, the broadband detector and digitizer or MHz are readily available and highly cost-effective. A comparison of those methods with PDI regarding the SNR performance, dynamics detecting capability is summarized in Supplementary Table 1. Importantly, our PDI method is able to capture all the harmonics and produce accurate thermal decay constant, which is beyond the reach of the multi-channel lock-in or the boxcar averaging approach.

When imaging specimens with background absorption, lock-in detection at the fundamental modulation frequency diminishes the contrast of fast decay signals. This is the case for detecting the small lipid droplets inside cells. These small organelles' signals are buried in the water background in the lock-in method, but are detectable with PDI and give higher contrast in higher-order harmonics. We note that this issue can be mitigated in a lock-in-based MIP microscope with tunable modulation frequency IR sources. If knowing the thermal dynamics of a particular type of absorber of interest, one can change the modulation frequency according to their transfer function for optimal detection. For those lipid droplets in water, which quickly decay in few hundreds of nanoseconds, modulation frequency at 500 kHz or higher is more suitable. Yet, the strategy of tuning modulation frequency is no longer valid if absorbers of different decay constants need to be imaged simultaneously.

By retrieving the thermal-physical properties, such as thermal diffusion in a specimen, IR camera-based nondestructive thermal imaging has enabled advanced applications for characterizing materials[51], including detection of hidden defects like cavities[52,53], points of failure in composites, solar cells, semiconductors, and electronics[54]. However, as the time scale comes to sub-microsecond and spatial resolution goes down to the nanoscale, such traditional temperature mapping methods have encountered difficulty due to largely reduced heat radiation and diffraction limit. Various novel techniques have been developed[55–60] to achieve transient heat detection in nanoscale. Our method intrinsically senses heat and is capable of measuring the exact temperature rise during the nanosecond photothermal process as demonstrated in Supplementary Note 3. This capability promotes PDI as a new scheme that directly probes those thermal processes at the far-field with nanometer-scale resolution and nanosecond temporal resolution.

Beyond the mid-IR photothermal process, PDI can fertilize research related to imaging of short lifetime events or complex photothermal decay process in general photothermal imaging field that includes but not limited to visible, near-infrared, and other transient physical perturbations[61,62]. Lock-in free PDI offers nanoseconds temporal resolution that is ultimately limited by the photon detector response. For example, it can be used to study the nonlinear photothermal phenomenon, including the nanobubble generation within a life span of hundreds of nanoseconds[63,64], higher-order thermal wave modulation on heat capacity and conductivity[65,66]; revealing resistive heating profile of nanostructure[8] and detecting photothermal induced pressure wave. Those photothermal processes have a relatively short lifetime due to the rapidly dissipated heat. On the other hand, PDI retrieves complete photothermal dynamics that reveal interesting heat dissipation processes of the sample for the first time. As shown in Supplementary Note 4, the lipid droplets inside a cell have a complex thermal decay composed of multiple distinct lifetimes. Such decay feature is related to the microenvironment of such small organelles conveying rich structure information to be scrutinized. This new imaging scheme can be applied to scrutinize intracellular organelles' thermal response together with chemical composition, or monitor transient cell response to fast perturbation[61], providing an entirely new perspective to understand the cell machinery. Collectively, the reported PDI system provides a novel far-field label-free imaging tool to scrutinize a sample's chemical composition and physical dynamics simultaneously.

## Methods

**Experimental setup.** The mid-IR PDI system (Fig. 1b) was built on an inverted microscope frame (IX71, Olympus). A pulsed mid-IR beam, generated by a tunable (from 900 cm$^{-1}$ to 1790 cm$^{-1}$) quantum cascade laser (MIRcat-2400, Daylight Solutions) is used as the pump source. The mid-IR beam passes through a Germanium dichroic mirror and then is focused onto the sample through a gold-coating reflective objective lens (52×; NA, 0.65; #66589, Edmund Optics). The residual mid-IR reflected by dichroic mirror is monitored with an MCT detector (PVM-10.6, VIGO System). The pulse repletion rate is tuned from 100 kHz to 1 MHz depending on the decay property of sample; the output pulse duration ranges from 300 to 1000 ns. Counter-propagated probe beam provided by a continuous-wave 532 nm laser (Samba 532 nm, Cobolt) is focused by a water immersion objective lens (×60; NA, 1.2; water immersion; UPlanSApo, Olympus). The probe beam is aligned to be collinear to the mid-IR pump beam to ensure the overlap of the two foci to maintain a good signal level. A scanning piezo stage (Nano-Bio 2200, Mad City Labs) with a minimum pixel dwell time of 50 μs and moving range of 200 μm is used to scan the sample. Backscattered probe photons are collected with a 50/50 beam splitter (BS013, Thorlabs), forward scattered probe photons are collected by the reflective objective and separated by the dichroic mirror. Both forward and backward probes are collected and sent to silicon photodiodes (S3994-01, Hamamatsu). The photocurrent is separated into AC and DC components with a bias Tee (ZFBT-4R2GW+, Minicircuit). The DC is a direct intensity readout. The AC channel is connected with a wideband voltage amplifier with 40 dB gain (DHPVA-101, FEMTO). The voltage signal is AC coupled and filtered with a low pass filter with a cut-off frequency of 25 MHz (BLP-25+, Minicircuit). Amplified signal is sent to a four-channel digitizer (Oscar 14, Gage Applied) installed on a computer, which has a sampling rate of 50 million samples per second and 14-bit A/D resolution. Acquired data is directly streamed to the computer memory for signal processing. During the imaging, the position feedback from scanning stage, mid-IR pulse shape from MCT are acquired synchronously by the same digitizer for image reconstruction and photothermal dynamics analysis. The same signal output from amplifier is also send to a lock-in amplifier (HF2LI, Zurich Instruments) for conventional MIP detection at the fundamental modulation frequency as comparison experiment and system calibration.

**Data acquisition and processing**. Acquired raw PDI raw data from the digitizer is directly streaming to the host computer memory via the PCIe Data Streaming Firmware (Gage Applied) and processed in the CPU. Segments containing repetitive measurements according to pixel dwell time are match filtered. Processed temporal dynamics with reduced data amount per pixel are then transferred to disk for image reconstruction.

During the match filtering processing, each segment is filtered in the frequency domain with a comb-like passband with each passing window at the harmonic of pump laser repetition rate (For laser running at 100 kHz, the pass windows are choosing at 100 kHz, 200 kHz, …, 2 MHz, 2.1 MHz). The window size is defined by the spectrum resolution according to pixel dwell time. The number of passing harmonics decides the thermal-dynamic bandwidth and influences the SNR. In this work, we used 16 order harmonics (1.6 MHz) for depicting absorption contrast which gives the highest image SNR. On the other hand, for depicting a complete photothermal dynamic profile, we used the bandwidth of 25 MHz. For best SNR performance, the included harmonics order can be selected based on the calculation model given in Supplementary Note 1S. for quantitative estimation.

With acquired temporal dynamic behind each pixel, the X-Y-t hyper image is then reconstructed according to the pixel position acquired from the scanning stage. For preserving positioning precision, we perform the reconstruction in an enlarged mesh with four times of acquired pixels by linear interpolation using ImageJ.

**Spectra acquisition**. The MIP spectra are acquired from PDI signal at different wavelengths and normalized according to the effective pulse energy. To minimize the heat dissipation influence and IR pulse duration variance, the absorption cross-section is evaluated at the initial 300 ns of the photothermal dynamic. The photothermal intensities at $t = 300$ ns at each wavelength are firstly evaluated noted as $S_{raw}(\lambda)$. Subsequently, the effective heating energy at each wavelength is evaluated by integrating the IR pulse shape from 0 ns to 300 ns, which is noted as $Q_{IR}(\lambda)$. With considering the IR focal spot size change along with wavelength, the final spectrum is then derived from normalizing $(S_{raw}(\lambda) * \lambda^2)/Q_{IR}(\lambda)$.

**Thermal decay fitting**. The thermal decay constant is acquired by fitting exponential function on the free decay process (IR pulse is finished) with a custom-coded Matlab program. The nonlinear regression model is used to retrieve the function with the formula: $y = y_0 + A_1 e^{-x/t}$.

The display decay constant map is acquired firstly by fitting the decay constant at every pixel in the PDI stack and then smoothed with a median filter.

**PMMA particles**. The PMMA particles with a nominate diameter of 300 nm (MMA300, Phosphorex) or with diameter of 500 nm (MMA500, Phosphorex) in solution form were first diluted with deionized water. Around 2 µL of the solution was then dropped on the surface of a calcium fluoride (CaF$_2$) substrate with 0.2 mm thickness for imaging. The photothermal signal was detected from backward scattering.

The 500-nm diameter PMMA particles (MMA500, Phosphorex) and 300-nm diameter PMMA particles (MMA500, Phosphorex) mixture sample was prepared by firstly mixing their solution together with 1:1 ratio, and then spreading one droplet of the solution on the surface of a CaF$_2$ substrate with 0.2 mm thickness for imaging. The photothermal signal is detected from backward scattering.

***E. coli***. Bacteria were harvested during the log phase. $^{12}$C or $^{13}$C-glucose was used as the only carbon source in the growth media. The growth media was removed from the bacterial samples by centrifugation and washing four times with 2 mL of deionized water. The bacterial pellet was then suspended in 0.25 mL of water, and 1 µL of the resulting bacterial suspension was dropped on CaF$_2$ with 0.2 mm thickness for imaging. The photothermal signal was detected from backward scattering.

**U87 cancer cells**. The U87 brain cancer cells were cultured on CaF$_2$ substrate for 24 h and fixed with formalin. The medium was washed three times by phosphate-buffered saline and replaced by 0.9% NaCl/D2O solution. It was later sandwiched with another thin calcium fluoride window with 0.2 mm thickness for imaging. The photothermal signal was detected from forward scattering.

**Reporting summary**. Further information on research design is available in the Nature Research Reporting Summary linked to this article.

## Data availability

All the data related to the work is available upon reasonable request to the corresponding author.

## Code availability

Data processing code related to the work is available upon reasonable request to the corresponding author.

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

## Acknowledgements

The work is supported by R35GM136223, R01AI141439, and R01CA224275 to J.X.C.

## Author contributions

L. L., J.Y., and J.X.C. conceived the idea. J.Y. and L.L. carried out modeling, PDI experiments, and analyzed the data. J.Y. and Y.Z. performed the PHI experiments. H.N. contributed to theoretical analysis. Y.T. prepared the cancer cells, M.Z. prepared the E. coli samples, Y.B. provided an initial demonstration of metabolism imaging in E. coli. J.Y. drafted the manuscript and all authors contributed to the manuscript writing. The authors thank Celalettin Yurdakul for the helpful discussion.

## Competing interests

J.X.C. declares a financial interest with Photothermal Spectroscopy Corp and Vibronix Inc. The rest of the authors declare no competing interest.
