## [Peer Review File · Nature Communications]

REVIEWER COMMENTS

Reviewer #1 (Remarks to the Author):

The manuscript given by Jiaze Yin and coauthors demonstrates a time-resolved dynamic photothermal microscopy technique with mid-infrared excitation. They present a lock-in-free detection of a point-scan-type photothermal microscope by continuous measurement of the signal with a wide bandwidth photodetector and digitizer. Using Fourier analysis, they demodulate not only the fundamental modulation-frequency signal but also higher harmonics so that extracting transient temporal information from the signal. The authors claim that the technique improves SNR by 4-fold to the lock-in counterpart and also provides effective separation in mid-infrared signal from water background by harnessing the different decay constant between the biomolecules and water.

It is a nice demonstration to show measuring the temporal behavior of the photothermal signal, but I feel that this technique is rather technically incremental to be published in Nature Communications that requires a significant conceptual or technical advancement. Taking temporal dynamics of ns-us time-scale is commonly used in various methodologies with off-the-shelf devices with a bandwidth of tens of MHz. The authors' demonstration might be new in the particular field of mid-infrared photothermal microscopy, which gives interest to this community. However, from a broader perspective, it is one of the demonstrations of this kind, and general interest and significance are missing. I would recommend it to be sent to journals dealing with more specialized materials. Detailed discussions about the contents are given below.

The SNR comparison between the authors' technique with harmonics and lock-in detection must be discussed more rigorously. In the supplementary materials (Supple. Fig. 4), the SNR is discussed, but it does not provide sufficient information to support the authors' claim saying 23 times improvement in SNR. What noise spectrum is assumed? What is time-constant (width of the bandpass filtering) of the lock-in detection assumed? How the time-constant affects the SNR improvement value? In the manuscript, 4-fold SNR improvement is claimed, but there is no quantitative analysis and explanation. Higher harmonics have lower SNR than the fundamental. How can one determine up to which harmonics should be added? A quantitative criterion should be discussed.

Taking phase from the lock-in detection also gives temporal decay information, which has been used quite often. I request the authors to discuss the comparison between the phase analysis and their techniques.

Detectors' and digitizers' specifications must be discussed for a fair comparison because one can use a relatively low-bandwidth detector and digitizer for lock-in detection only. On the other hand, the authors' technique requires a high-bandwidth detector/digitizer. In the manuscript, the authors assume using the same detector for both the methods, but, in reality, lock-in users can use an optimal one. More discussion is required.

In Figure 2f, There are apparent differences between the FTIR and MIP spectra, although the authors mention that they are in good agreement in the manuscript. What is the potential cause of the discrepancy?

In Equation 3, does Newton's law work in this time range? The assumption of the theory must be reviewed.

The water signal is observed at the lipid droplet in the cells' images. Why is that? One can imagine that there is less water where lipid droplet is localized. It should be discussed.

Reviewer #2 (Remarks to the Author):

Yin et al. present a nice addition to their already impressive work on mid-infrared (MIR) photothermal microscopy. To me, the main contribution of their manuscript "Nanosecond-Resolution Photothermal Dynamic Imaging via MHz Digitization and Match Filtering" is the thorough discussion of lock-in amplifier-based signal demodulation in the limit of non-sinusoidal signal modulation. Albeit being rather technical, this work is an important contribution as traditional photothermal microscopy crucially relied on AOM-based signal modulation for which lock-in detection is ideally suited. The rapidly developing, and highly-promising, field MIR-photothermal imaging, on the other hand, necessary relies on near-square wave excitation and alternative detection approaches are therefore needed. Yin et al. clearly outline such a strategy and their implementation is certainly going to be adopted by other groups in their as well as related fields. As such, I recommend publishing the manuscript in Nature Communications following some minor corrections.

In more details,

1) My interpretation of the manuscript is that the authors implement a Fourier-transform based boxcar integration method, with slightly weighted harmonics, that operates on digitised data. As such, it would be nice if the authors could discuss their advance with respect to both boxcar and lock-in technology and not just with respect to first-generation lock-in amplifiers. Even though the latter have been the go-to technology for historical implementations of photothermal microscopy boxcar-based signal detection is an established methodology and state-of-the-art lock-in amplifiers often offer both modalities as well as multi-harmonics demodulation (See for example UHFLI from Zurich Instruments). The UHFLI allows two-channel lock-in with 4 harmonics per channel thus detecting 8 harmonics without the need for custom electronics and complex signal analysis. As such, the massive signal-to-noise improvements quoted by the authors might be correct, when compared to their commercial device of choice, but I would attribute most of the increase to a poor choice of hardware rather than a groundbreaking new signal demodulation approach. As such, I would recommend a more cautious approach to presenting this aspect of their work.

2) The author digitise all data and then perform signal-analysis post acquisition. This approach enabled recording, and analysing, the nice thermalisation dynamics reported throughout the paper. While certainly educative, and fully justified in the context of this paper, such an approach can be difficult to implement for large scale imaging as it generates large amounts of data that probably take longer to analyse than to acquire (around 10 GB/min I would guess). As an alternative, a simple FPGA-based boxcar integrator based on a few boxes such as: pre-excitation, a 200-500 ns and a 2000-3000 ns box should yield near-identical information, with identical SNRs, but in real-time and without generating vast amounts of data. Have the authors considered such an alternative?

3) It would be nice if the authors could use relative numbers (mV change vs V detected) to represent their photothermal signals rather than using absolute numbers. As such, it is easier for other groups to put the numbers into a relevant experimental context.

4) I don't understand why the authors measure zero raw photodiode signal if not MIR excitation is present (Figure 1d). The authors don't mention any high pass filters and I would have expected a signal proportional to the photon flux.

5) The data shown in Figure 2g seems to exhibit a signal gradient (top to bottom). I would expect near-identical signals for a raster-scanning based technique that is not limited by the spatial extend of the pulses. What is the reason for this gradient?

6) I don't understand how it is possible to conclude that the sample doesn't overheat (Figure 3) based on the decay constant of a backscattering-detected signal. In my understanding, bulk water heating should predominantly manifest itself in the signal detected in the forward direction. It would be nice if the authors could give some rough temperature-increase estimates. 35 mW absorbed in a 5-10 μm spot seems like a huge amount of power. Figure 4 underlines this problem where the decay constant for water-thermalisation is quantified to being on the order of 5 μs , as such, I expect that the 1 MHz experiments performed on bacteria might considerably impact the bacterias' metabolism by considerably altering the steady-state temperature of the sample. Are the powers chosen within the biologically feasible range?

7) Figure 4: what are the units of the z-axis?

8) I really like the idea of quantifying the decay constants (Figure 4) and it would be fantastic if it would be possible to better visualise the data. Maybe the authors could consider only showing decay constants for regions with larger signals (I am mainly refereeing to Figure 4f)? As such, it

would be possible to appreciate the differences in lipid droplet decay constants, and maybe see droplet-size vs decay relations. Currently, this is, unfortunately, impossible as the uniform water background around 5 μ s (if I interpret the z-scale correctly) turns the, hopefully, very informative image into a red-blue two-colour image where all the interesting details are lost.

9) Figure 5a,b please set both log scales to the same minimum as the offset in (a) is masking the noise.

Reviewer #3 (Remarks to the Author):

Yin. et al. report in their manuscript entitled "Nanosecond-Resolution Photothermal Dynamic Imaging via MHz Digitization and Match Filtering" about a new experimental analysis technique for infrared photothermal microscopy, allowing them to filter the signal generated by the impulsive heating of objects by their temporal response. The temporal response is largely governed by heat capacity and thermal conductivity. Therefore, analyzing the temporal signal provides information on the thermal characteristics of the sample. Furthermore, the temporal decay of the signal is used to discriminate different origins of the signal (e.g. water background or lipids). The method is demonstrated on several types of samples, including biological samples.

This is a very nice approach to the analysis of photothermal signals. Moreover, it is a straightforward extension of time-resolved detection in other fields of science, which is demonstrated to be very useful in the context of photothermal detection. I, therefore, can recommend the paper for publication after the authors have commented/discussed the following additional issues:

1) The method analyzes an exponential decay in the simplest case, while later, more complex signal decays/rises are used to separate background (water) signals from other contributions. I would like the authors to discuss the expected temporal response of the sample in the case of a mixed system (e.g. lipid droplets in water) in the model section more clearly. I think it is important for the reader to understand how the temporal response of a more complex system arises.

2) Complex decays after impulsive excitation are analyzed in many fields of science, e.g. fluorescence microscopy. It would be nice if the authors refer to such approaches in the model section.

3) For complex temporal decays, I would usually expect an analysis by a Laplace transform or some maximum entropy method that enables direct extraction of timescales. Fourier approaches usually give the same info in terms of Lorentzians. Could the authors discuss their approach in terms of these transformations, which would readily connect to different fields of science?

3) The IR powers that are incident to the samples are on the order of 10 to several 10 mW. Is that a time average over many pulses? What is the input power per pulse, and what is the temperature perturbation, especially in the biological samples?

Reviewer #1 (Remarks to the Author):

The manuscript given by Jiaze Yin and coauthors demonstrates a time-resolved dynamic photothermal microscopy technique with mid-infrared excitation. They present a lock-in-free detection of a point-scan-type photothermal microscope by continuous measurement of the signal with a wide bandwidth photodetector and digitizer. Using Fourier analysis, they demodulate not only the fundamental modulation-frequency signal but also higher harmonics so that extracting transient temporal information from the signal. The authors claim that the technique improves SNR by 4-fold to the lock-in counterpart and also provides effective separation in mid-infrared signal from water background by harnessing the different decay constant between the biomolecules and water.

1. It is a nice demonstration to show measuring the temporal behavior of the photothermal signal, but I feel that this technique is rather technically incremental to be published in Nature Communications that requires a significant conceptual or technical advancement. Taking temporal dynamics of ns-us time-scale is commonly used in various methodologies with off-the-shelf devices with a bandwidth of tens of MHz. The authors' demonstration might be new in the particular field of mid-infrared photothermal microscopy, which gives interest to this community. However, from a broader perspective, it is one of the demonstrations of this kind, and general interest and significance are missing. I would recommend it to be sent to journals dealing with more specialized materials. Detailed discussions about the contents are given below.

Re: We thank the reviewer's thoughtful comments. A more detailed discussion regarding the significance of this technique is addressed from 5 aspects below:

1) The mid-infrared photothermal (MIP) imaging field is an emerging and promising field as a valuable tool for biological and material science, which is well elaborated in a recent review (Science Advances 7(20), 2021) ¹. It enables submicron mid-IR imaging for a broad spectrum of applications in both research and industry, including but not limited to failure analysis, single bacterium antibiotic test, neural amyloid aggregation analysis^{2,3}, and high-resolution IR histology. Since the first demonstration of high-performance MIP imaging in 2016 (Science Advances 2(9) 2016)⁴, it has been quickly commercialized into a product by Anasys (now Photothermal Spectroscopy Corp). There has been an exponential increase in the number of publications based on MIP microscopy. Yet, the current MIP microscope does not provide thermal dynamics information, which is a key parameter to probe the environmental properties. The current work addresses this significant issue.

2) This method is not limited to mid-infrared photothermal imaging but is also applicable to photothermal imaging at other wavelengths including visible photothermal imaging. The PDI method will empower the general photothermal imaging field with the added thermal dynamics information.

3) As the time scale comes to sub-micron second and spatial resolution goes down to the nanoscale, traditional IR camera-based thermal imaging methods have encountered difficulties due to largely reduced heat radiation and diffraction limit. Transient heat detection in nanoscale is another important hot topic with various technologies that have been developed⁵⁻⁷. Our method intrinsically senses heat is capable of measuring the exact temperature rise during the nanosecond photothermal process. PDI is a new scheme that directly links to those thermal field research with nanometer spatial resolution and nanosecond temporal resolution from far-field sensing.

4) Heterodyne is a major method-used for photothermal detection. We break the convention here. The lock-in detection is not a universal optimal method to reach high sensitivity measurement in every photothermal imaging setup. Leveraging the widely accessible broadband photodetector and digitizer, the general photothermal imaging community can apply this simple implementation and gain two-fold benefits in both sensitivity enhancement and retrieving complete thermal dynamics for their specific applications.

5) The PDI can be further extended to enable ultrafast photothermal imaging. We solve one fundamental limitation that existed for lock-in based photothermal imaging, where multichannel is difficult to achieve. There are limited solutions of multiplex lock-in detection, and the current state-of-the-art device (UHFLI, Zurich) can only reach to 2 channels. In contrast, high-speed digitizers with hundreds of channels are mature modules used in other imaging fields, such as ultrasound imaging. Typical ultrasound can acquire 128 channels at a sampling rate of 65 Mega-Sample/second (Vantage 128, Verasonics). We envision another two orders of magnitude speed improvement over our current system by utilizing spatial multiplexed PDI detection schemes such as line scanning.

In summary, PDI that uses widely accessible broadband photodiode and digitizer offers a new paradigm for photothermal imaging with enhanced sensitivity and complete thermal dynamic information. It represents a significant advance and will have a broad impact. It benefits not only the mid-infrared photothermal imaging but also the general thermal imaging research and studies. In the revised manuscript, paragraphs discussing the points above are added in the introduction on page 3 and the discussion on page 16.

2. The SNR comparison between the authors' technique with harmonics and lock-in detection must be discussed more rigorously. In the supplementary materials (Supple. Fig. 4), the SNR is discussed, but it does not provide sufficient information to support the authors' claim saying 23 times improvement in SNR. What noise spectrum is assumed? What is time-constant (width of the bandpass filtering) of the lock-in detection assumed? How the time-constant affects the SNR improvement value? In the manuscript, 4-fold SNR improvement is claimed, but there is no quantitative analysis and explanation. Higher harmonics have lower SNR than the fundamental. How can

one determine up to which harmonics should be added? A quantitative criterion should be discussed.

Re: We deeply appreciate the referee’s constructive comments. A more rigorous and detailed discussion of the SNR improvement is provided as follows:

- 1) The improved SNR in PDI comes from the following two aspects. Firstly, under the pulse excitation, harmonics at higher-order have comparable or even higher SNR than fundamental frequency, for laser noise is largely reduced at a higher frequency while signal diminishes slowly. Secondly, harmonics of the signal are correlated and can add up while the uncorrelated random noise can't.
- 2) The noise spectrum in our system is shown in **Fig. R1**, which was measured by a broadband photodiode using the lock-in sweeper function when the IR laser was off (laser noise) and visible laser off (electrical noise). The laser noise density follows a $1/f^\alpha$ distribution in the low frequency (up to 1 MHz), and it reduces to white noise after that. Given the thermal decay of sub-microsecond to microsecond, the photothermal signal existed at the sub-MHz level. Thus, the laser $1/f$ noise is the dominant noise in conventional photothermal imaging with low modulation frequency, and lock-in detection cannot eliminate completely by using a long integration time. Techniques like balanced detection are required.

Figure R1. Experimental measured system noise spectrum.

- 3) The quantitative analysis for SNR improvement times is given below. Taking the analogy to first order resistor-capacitor circuit, the photothermal signal under impulse excitation has a frequency response $H(f)$ given by:

$$H(f) = \frac{1}{1 + j2\pi\tau f}$$

Therefore, for an absorber with defined decay constant, its photothermal signal amplitude S under pulse excitation with n^{th} correlated harmonics captured is given by the following relation, where A is a scaling factor:

$$S = S_1 + S_2 + \dots + S_n = A(|H(f_1)| + |H(f_2)| + \dots + |H(f_n)|)$$

The uncorrelated noise amplitude N captured in the same n^{th} harmonics is calculated by:

$$N = \sqrt{|N_1|^2 + |N_2|^2 + \dots + |N_n|^2},$$

where N_n is the noise amplitude at n^{th} harmonics frequency. In addition, the amplitude of noise has the relationship of $N_i^2/N_1^2 = f_1/f_i$ for $1/f$ noise dominant region.

The general SNR is then given by S/N . We note that the SNR_{LIA} for lock-in detection of the first order harmonic is given by S_1/N_1 , and define the amplitude's ratio between i^{th} harmonic and first harmonic of signal $a_i = S_i/S_1$, and of noise $b_i = N_i/N_1$ correspondingly. SNR of PDI with n^{th} concurrently detected harmonics is then written as:

$$SNR_{PDI}(n) = \frac{(1 + a_2 + \dots + a_n)}{\sqrt{1 + b_2^2 + \dots + b_n^2}} SNR_{LIA}$$

The SNR improvement is subject to the term $(1 + a_2 + \dots + a_n)/\sqrt{1 + b_2^2 + \dots + b_n^2}$. Based on the above equation, we estimate the SNR improvement factor for photothermal imaging of $D=300$ nm PMMA particle with a decay constant of 280 ns. With IR repetition rate at 100 kHz and 20 kHz respectively. The estimated PDI SNR improvement times versus captured harmonics order is shown in **Fig.R2**. By capturing up to 16 order of harmonics (1.6 MHz), 5.4 times SNR improvement will be expected for IR excitation at 100 kHz (**Fig.R2a**). Experimentally, our result shows a 4.3 times improvement, which is close. The discrepancy might be attributed to the non-ideal pulse shape of the actual IR excitation pulse while an ideal IR impulse excitation is assumed and reduced $1/f$ noise at megahertz frequency. For IR excitation at 20 kHz, a gain of 22 times improvement is expected by captured all the harmonics within 1.6 MHz (**Fig.R2b**).

We add these discussions in the result section on page 7 in the revised manuscript. A more detailed quantitative analysis of the SNR in comparison to the lock-in method is revised in the **Supplementary S.1**.

Figure R2. Estimated SNR improvement times versus captured harmonics order. (a) SNR improvement factor of photothermal detection D=300 nm PMMA particle with decay constant of 280 ns under IR repetition rate of 100 kHz. **(b)** SNR improvement factor of detecting the same sample of (a) under IR repetition rate of 20 kHz. Impulse excitation and 1/f noise model are assumed for the above results.

3. Taking phase from the lock-in detection also gives temporal decay information, which has been used quite often. I request the authors to discuss the comparison between the phase analysis and their techniques.

Re: Thanks for the referee's kind suggestion. A more detailed comparison of temporal decay retrieved from the lock-in phase and PDI detection is given below:

- 1) Lock-in phase readout at reference frequency can also be used to quantitatively retrieve the time constant of the decay process. However, it is only suitable for retrieving the decay signal with a well-defined model. It is also limited in accurately assessing the complex thermal dynamic process. For example, the phase information is complicated if there are two decay processes superposed together⁸. Secondly, an accurately measured phase delay requires either a sinusoidal excitation or an ideal impulse response, which is practically difficult for mid-infrared photothermal imaging. Moreover, in terms of SNR, the phase method has poor accuracy for low SNR data and it is susceptible to error from instrument response⁹. Instead, PDI, by detecting the complete temporal dynamics, is universal for quantitative analysis of the photothermal process and it can reveal unknown phenomenon.

2) In the single decay model, the phase value to the decay constant has a non-linear relation and frequency dependency¹⁰. It results in a limited dynamic range in telling the thermal property difference of materials inside. As shown in **Fig.R3**, a phase difference of roughly two degrees can represent large different decay constant changes at the demodulation frequency of 100 kHz. Moreover, the phase value and contrast will change over the modulation frequency, making them hard to interpret. On the other hand, PDI retrieved information is free of issues mentioned above.

Figure R3. Phase delay versus modulation frequency of absorbers with different thermal decay constants.

Collectively, the traditional lock-in phase method is applicable to scrutinize decay property for well-defined models or samples, yet, it has limited accuracy in assessing complex systems and its non-linear response, frequency dependency complicate the data interpretation. On the contrary, the PDI technique has a complete signal profile and allows comprehensive analysis. The comparison above is added in the introduction section on page 3 of the revised manuscript and highlighted in blue.

4. Detectors' and digitizers' specifications must be discussed for a fair comparison because one can use a relatively low-bandwidth detector and digitizer for lock-in detection only. On the other hand, the authors' technique requires a high-bandwidth detector/digitizer. In the manuscript, the authors assume using the same detector for both the methods, but, in reality, lock-in users can use an optimal one. More discussion is required.

Re: We thank the reviewer for raising this point. A more detailed comparison between the lock-in and PDI method is below:

1) For the digitizer requirements:

The lock-in equipment demodulates the raw signals to low-speed signals. It works with an extra low bandwidth digitizer to digitize the signal and send it to the PC. But the lock-in equipment used in the current state-of-the-art photothermal imaging system actually employs an internal high-speed digitizer to do the demodulation. For

example, the Zurich UHFLI lock-in is equipped with a 1.8 GS/s digitizer, which is even higher than 50MS/s digitizer PDI uses. Moreover, the Zurich lock-in costs \$200k, while the PDI digitizer is only \$6k. The PDI method can be more cost-effective and widely accessible while offering higher SNR.

2) For the photodetector requirements:

Indeed, a low bandwidth photodiode can be used in the conventional lock-in method. But a low bandwidth photodiode will be slow to capture the transient photothermal dynamics that happen rapidly. Also, the broadband photodetector is widely accessible. We chose a commonly used off-shelf photodiode with 25 MHz bandwidth from Hamamatsu, and a much higher speed photodiode with over GHz bandwidth is also readily available in optical communications.

Thus, there is essentially no significant difference in the bandwidth demand of digitizers between the lock-in and PDI methods. The broadband photodetector used in PDI is widely accessible and it allows the capture of fast thermal dynamics to scrutinize the samples inside.

5. In Figure2f, There are apparent differences between the FTIR and MIP spectra, although the authors mention that they are in good agreement in the manuscript. What is the potential cause of the discrepancy?

Re: We thank the reviewer's keen observation! We improved the normalization of IR power and the new spectrum data is improved, as shown in **Fig. R4a**.

We improved the normalization by two means:

- 1) Taking into account the IR focus size variation at different IR wavelengths rather than directly using the power measured by the IR detector for normalization. Given the IR focus size is proportional to the wavelength, the previous IR spectrum is further normalized by dividing with $1/\lambda^2$.
- 2) Considering the variation of the spectral response by the IR power detector at different wavenumbers (**Fig. R4b**).

With the above corrections, the new spectrum shows better consistency with the FTIR spectrum in peak ratios now, and it is updated in the revised manuscript. The new normalization scheme is updated method section.

Figure R4. (a) Mid-infrared spectrum of PMMA. (b) Mid-IR power detector response spectrum.

6. In Equation 3, does Newton's law work in this time range? The assumption of the theory must be reviewed.

Re: We thank the reviewer for this kind suggestion. We reviewed the heat transfer model and applicable conditions. We now refined the heat dissipation equation with the assumption on page 5 of the manuscript, that: "*The heat transfer model here applies when the time scale of the heat dissipation is longer than sub-nanosecond.*" As discussed in detail by Chen et al, Newton's law of cooling is feasible for the nanosecond scale photothermal process. During the photothermal process, the heat dissipation happened around 100 picoseconds or longer, the heat transfer model no longer holds for the heating pulse with a duration shorter than that. In our scenario, the heat transfer happened in nano to microsecond scale, Newton's law is still applicable in our thermal analysis.

7. The water signal is observed at the lipid droplet in the cells' images. Why is that? One can imagine that there is less water where lipid droplet is localized. It should be discussed.

Re: We thank the reviewer for raising this point. The water signal observed at the lipid droplet is a weak signal from water. Photothermal contrast originates from the scattering field modulation, which is directly related to the refractive index mismatch ($n_s - n_m$), where n_s and n_m are the refractive indexes of lipid droplets and medium. Due to the IR absorption, n_m of water experiences a modulation, and it alters the refractive index mismatch ($n_s - n_m$) by heat, which shows up as photothermal contrast. As compared with pure water background, this particular signal from water is amplified by the lipid droplet that has a large scattering bias.

Reviewer #2 (Remarks to the Author):

Yin et al. present a nice addition to their already impressive work on mid-infrared (MIR) photothermal microscopy. To me, the main contribution of their manuscript "Nanosecond-Resolution Photothermal Dynamic Imaging via MHz Digitization and Match Filtering" is the thorough discussion of lock-in amplifier-based signal demodulation in the limit of non-sinusoidal signal modulation. Albeit being rather technical, this work is an important contribution as traditional photothermal microscopy crucially relied on AOM-based signal modulation for which lock-in detection is ideally suited. The rapidly developing, and highly-promising, field MIR-photothermal imaging, on the other hand, necessary relies on near-square wave excitation and alternative detection approaches are therefore needed. Yin et al. clearly outline such a strategy and their implementation is certainly going to be adopted by other groups in their as well as related fields. As such, I recommend publishing the manuscript in Nature Communications following some minor corrections.

In more details,

1) My interpretation of the manuscript is that the authors implement a Fourier-transform based boxcar integration method, with slightly weighted harmonics, that operates on digitised data. As such, it would be nice if the authors could discuss their advance with respect to both boxcar and lock-in technology and not just with respect to first-generation lock-in amplifiers. Even though the latter have been the go-to technology for historical implementations of photothermal microscopy boxcar-based signal detection is an established methodology and state-of-the-art lock-in amplifiers often offer both modalities as well as multi-harmonics demodulation (See for example UHFLI from Zurich Instruments). The UHFLI allows two-channel lock-in with 4 harmonics per channel thus detecting 8 harmonics without the need for custom electronics and complex signal analysis. As such, the massive signal-to-noise improvements quoted by the authors might be correct, when compared to their commercial device of choice, but I would attribute most of the increase to a poor choice of hardware rather than a groundbreaking new signal demodulation approach. As such, I would recommend a more cautious approach to presenting this aspect of their work.

Re: We thank the reviewer’s constructive feedback and efforts towards improving our manuscript. In the following, we summarize and compare the mainstream signal demodulation instrument and their performance for extracting temporal dynamics information in **Table R1** below.

	Lock-in	Boxcar	PDI
Output contrast	Amplitude at selected harmonics; up to 8 channels	Amplitude difference between on-off state	Complete temporal dynamics of modulation
Dynamics information	Phase delay to the reference; up to 8 channels	Temporal dynamics can be acquired by scanning the gating delay	Comprehensive decay information acquired at once
Preferred input signal for high SNR	50% duty cycle	Low duty cycle (<50%)	No requirements, versatile processing methods to match different signals
User end signal processing	Not required	Not required	Required
Input bandwidth	600MHz (1.8GHz digitizer inside)	600MHz (1.8GHz digitizer inside)	Half of digitizer sampling rate (Up to 2GHz)
Maximum channel number	2-channel	2-channel	128 channel or more for high freedom in customization

Table R1: Comparison of state of art lock-in, boxcar and PDI for temporal dynamics detection

We agree with the reviewer that digital boxcar integrator or multi-harmonics correlation from lock-in may provide a similar level of SNR improvement of PDI for detecting photothermal signals under pulsed excitation. Our technology has a key advancement that is revealing comprehensive photothermal dynamics with high temporal resolution. Such information would require further model fitting and Fourier synthesis for multi-harmonics demodulation^{11,12} using lock-in method or thousands of repetitive and tedious measurements of tuning the gating window delay using boxcar.

In addition, we solve the fundamental issue of limited channel numbers for multiplexed detection in the lock-in amplifier. Being not limited to two channels, our lock-in free method allows spatial multiplex detection up to hundreds of channels with the off-shelf instrument and it can largely improve the photothermal detection throughput.

These discussions above are now included in paragraph 2 in the discussion section of the revised manuscript.

2) The author digitise all data and then perform signal-analysis post acquisition. This approach enabled recording, and analysing, the nice thermalisation dynamics reported throughout the paper. While certainly educative, and fully justified in the context of this paper, such an approach can be difficult to implement for large scale imaging as it generates large amounts of data that probably take longer to analyse than to acquire (around 10 GB/min I would guess). As an alternative, a simple FPGA-based boxcar integrator based on a few boxes such as: pre-excitation, a 200-500 ns and a 2000-3000 ns box should yield near-identical information, with identical SNRs, but in real-time and without generating vast amounts of data. Have the authors considered such an alternative?

Re: We thank the reviewer for raising this question. The FPGA-based boxcar can be a good alternative to achieve similar SNR by doing on- and off- differential detection, but it will require a tedious sweep of delay window to acquire the entire thermal decay trace. On the contrary, the PDI acquires temporal thermal dynamics at once, which is simpler.

As regards the data size, the raw data acquired in our system is 6GB/min, which is relatively large. But practically, this is not a limiting factor for applications. We solved this issue by taking advantage of the direct memory access (DMA) function provided by the digitizer manufacture. Raw data is stored temporally in memory and processed immediately during acquisition. In our experiments, it is typically reduced to roughly 1GB/min for imaging with a pixel dwell time of 100 μ s. We have updated our solution in the method part of the revised manuscript.

Lastly, this raw data throughput of a few GB/min is common and acceptable for current imaging setups. For example, optical coherence tomography typically outputs data at a rate over 60GS/min (2K samples per Aline, 0.5 MHz A-line rate).

3) It would be nice if the authors could use relative numbers (mV change vs V detected)

to represent their photothermal signals rather than using absolute numbers. As such, it is easier for other groups to put the numbers into a relevant experimental context.

Re: We thank the good suggestion! The change has been made in the main text.

4) I don't understand why the authors measure zero raw photodiode signal if not MIR excitation is present (Figure 1d). The authors don't mention any high pass filters and I would have expected a signal proportional to the photon flux.

Re: We thank the reviewer to point out this question. We have added the details in methods. In our system, the photodiode output current is firstly converting to voltage with a load resistor and separated into AC/DC components with a bias T (equivalent to high pass filter >10kHz). Only the AC signal is amplified, and the photon flux is represented in the DC channel. The amplifier and digitizer are all working in AC coupling mode (>10Hz). The AC coupling arrangement in our system allows a higher dynamic range to resolve the weak modulation signal on a large background. We updated these details in the methods section of the revised manuscript.

5) The data shown in Figure 2g seems to exhibit a signal gradient (top to bottom). I would expect near-identical signals for a raster-scanning based technique that is not limited by the spatial extend of the pulses. What is the reason for this gradient?

Re: We thank the reviewer's keen observation. After checking the data, a small mean-variance of 10% in intensity along with a slight difference in point spread function existed in the top and bottom boundary of this image. This is potentially caused by the sample slide tilting in the large field of view (~150 μm between the top and down). These minor issues can potentially be mitigated by using an autofocus module or calibration over a large uniform sample.

6) I don't understand how it is possible to conclude that the sample doesn't overheat (Figure 3) based on the decay constant of a backscattering-detected signal. In my understanding, bulk water heating should predominantly manifest itself in the signal detected in the forward direction. It would be nice if the authors could give some rough temperature-increase estimates. 35 mW absorbed in a 5-10 μm spot seems like a huge amount of power. Figure 4 underlines this problem where the decay constant for water-thermalisation is quantified to being on the order of 5 μs , as such, I expect that the 1 MHz experiments performed on bacteria might considerably impact the bacterias' metabolism by considerably altering the steady-state temperature of the sample. Are the powers chosen within the biologically feasible range?

Re: We appreciate the reviewer's point of biosafety in our imaging methods. In short, the peak temperature rising of bacterium cell per heating pulse is less than 10 Kelvins, which is biologically safe. A detailed discussion is given below:

- 1) The bacterium sample was drop cast on a coverslip without buffer medium but surrounded with air. The detected signal originates from the bacterium's intrinsic backscattering change due to its intrinsic temperature rise.
- 2) We estimate the temperature rising with the corresponding modulation depth according to our AC/DC signal amplitude. Under an IR excitation with 980 ns pulse width, the photothermal signal from bacteria cells have a peak amplitude A_s around 10mV. The DC intensity channel A_{DC} output a voltage of 9 mV. The modulation depth is then derived by $A_s/(Gain * A_{DC})$, where the gain used was 40 dB (100X). The calculated modulation depth is about 1.1%. With a conservative estimation, the sample scattering intensity change is $10^{-4}/K$ ¹³. A transient 11 Kelvins rising is estimated, and such temperature only maintains for less than one hundred nanoseconds. For the 1 MHz modulation, the IR pulse reduces to 300 ns with same peak intensity, the modulation depth reduced to 0.3%, the transient temperature rise is estimated as 3 Kelvins.

7) Figure 4: what are the units of the z-axis?

Re: We now change the units to mV as amplitude unit.

8) I really like the idea of quantifying the decay constants (Figure 4) and it would be fantastic if it would be possible to better visualise the data. Maybe the authors could consider only showing decay constants for regions with larger signals (I am mainly refereeing to Figure 4f)? As such, it would be possible to appreciate the differences in lipid droplet decay constants, and maybe see droplet-size vs decay relations. Currently, this is, unfortunately, impossible as the uniform water background around 5 μ s (if I interpret the z-scale correctly) turns the, hopefully, very informative image into a red-blue two-colour image where all the interesting details are lost.

Re: We thank the reviewer's suggestion, now the image for the lipid channel is plotted from 100 ns to 1 μ s range. One significant observation is for single lipid droplets always hold a swift decay less than 500 ns, while for the cluster of lipids, the decay is slower.

9) Figure 5a,b please set both log scales to the same minimum as the offset in (a) is masking the noise.

Re: We thank the reviewer's kind suggestion, the offset are now plotted with the minimum offset.

Reviewer #3 (Remarks to the Author):

Yin. et al. report in their manuscript entitled "Nanosecond-Resolution Photothermal Dynamic Imaging via MHz Digitization and Match Filtering" about a new experimental analysis technique for infrared photothermal microscopy, allowing them to filter the signal generated by the impulsive heating of objects by their temporal response. The

temporal response is largely governed by heat capacity and thermal conductivity. Therefore, analyzing the temporal signal provides information on the thermal characteristics of the sample. Furthermore, the temporal decay of the signal is used to discriminate different origins of the signal (e.g. water background or lipids). The method is demonstrated on several types of samples, including biological samples.

This is a very nice approach to the analysis of photothermal signals. Moreover, it is a straightforward extension of time-resolved detection in other fields of science, which is demonstrated to be very useful in the context of photothermal detection. I, therefore, can recommend the paper for publication after the authors have commented/discussed the following additional issues:

1) The method analyzes an exponential decay in the simplest case, while later, more complex signal decays/rises are used to separate background (water) signals from other contributions. I would like the authors to discuss the expected temporal response of the sample in the case of a mixed system (e.g. lipid droplets in water) in the model section more clearly. I think it is important for the reader to understand how the temporal response of a more complex system arises.

Re: We appreciate the reviewer's good suggestion. For proof of concept, the transient heat conduction model we introduced is a simple case. It gives an accurate description of the thermal response for those nano-size absorbers and helps separate the water background. We agree with the reviewer that temporal response can be complex depending on the specific sample and microenvironments, and complex thermal modeling will help.

Here, we use the experimentally measured complex decay signal from lipid droplets inside a cell as an example. As shown in **Fig.R5**, and this discussion has been added in the supplementary session S.4. As shown in **Fig.R5**, the heat dissipation process of lipid droplets inside a cell is a superposition of two decay processes with a different lifetime. These multiple lifetimes reveal interesting facts that: in a complex system, the heat dissipation power Q_{diss} of those absorbers is driven by $hS[T(t) - T_{\text{env}}(t)]$, in which the temperature of its microenvironment $T_{\text{env}}(t)$ can no longer be simplified as a constant value, but a time-dependent variable. Those PDI detected interesting thermodynamics reveal highly diverse thermal properties in nanoscale, which contains wealth information of the cells' structure.

Figure.R5. Complex decay signal with multiple time constant and their position in cell.

The complex decay example above is now added in **Supplementary S.4** and **Supplementary Figure 6** of the supplementary document.

2) Complex decays after impulsive excitation are analyzed in many fields of science, e.g. fluorescence microscopy. It would be nice if the authors refer to such approaches in the model section.

Re: We appreciate the reviewer's good suggestion. The analysis methods of complex decays are well developed and studied in many fields, such as fluorescence lifetime microscopy. Leveraging these advanced methods and models will definitely boost the exploitation of the rich thermal dynamic information offered by the PDI method. We add those studies of the complex decays in the model section on page 5 with the following note: "Note that a single exponential decay model is employed for illustration. For complex sample configurations, where heat conduction inside absorber or micro-environment needed to be considered, the assumption that constant ambient temperature T_{env} no longer holds. The transient photothermal dynamics have superimposed decay with multiple lifetimes. For those cases, models that study complex decays in many other fields, such as fluorescent lifetime imaging, can be applied."

3) For complex temporal decays, I would usually expect an analysis by a Laplace transform or some maximum entropy method that enables direct extraction of timescales. Fourier approaches usually give the same info in terms of Lorentzians. Could the authors discuss their approach in terms of these transformations, which would readily connect to different fields of science?

Re: We deeply appreciate the reviewer's recommendation of those advanced temporal dynamics analysis methods. For this work, we majorly demonstrate the technique as a proof-of-concept study of acquiring complete dynamics information and using it to suppress the water background. Regarding the model estimation, we used a least-square fitting method to minimize the Gaussian-distributed noise, which is not optimized and not suitable for multiple lifetime analysis.

We add a note in the results part on page 8 of the manuscript when we firstly introduce the fitting analysis to inform the readers that specific and advanced transform or methods can be used to model the thermal dynamics, besides using the basic fitting approaches. *"Here we extracted the exponential decay constant with least square fitting method. Note that advanced methods used including maximum likelihood estimation, maximum entropy methods can be exploited to obtain the decay information, as shown in many other time-resolved spectroscopy field of studying complex decays."*

3) The IR powers that are incident to the samples are on the order of 10 to several 10 mW. Is that a time average over many pulses? What is the input power per pulse, and what is the temperature perturbation, especially in the biological samples?

Re: We appreciate the reviewer raising this temperature rise issue. Our point-to-point answers are as follows:

- 1) The IR laser in our experiments ran in pulsed mode with a pulse width of 0.3 to 0.98 μs operating at 100 kHz to 1 MHz. The incident power of mW in the main text is the averaged power of IR laser measured at the input port of the focusing objective lens in the above configurations. We now clarify this in the revised manuscript.
- 2) The input pulse energy at the laser output is 30 to 100 nJ for pulse width of 980 ns, dependent on the wavelength selected.
- 3) For an accurate estimation of the temperature rise, we take the 500 nm PMMA particle as an example. It was estimated to be 7.6 K, which is biologically safe. Below is the calculation process.

Here, we retrieved the temperature rising from D=500 nm PMMA particle as shown in **Fig. R6**. The MIP signal from the PMMA particle originates from the scattering intensity modulation as the size and refractive index change due to heating. Such scattering field change per kelvin can be evaluated by utilizing the Mie-scattering theory for given material and size as introduced in reference¹⁴. For the backscattering of D=500 nm

PMMA particle with a collection NA of 1.2, the average change of scattering intensity is 0.39%/K. With the raw PDI signal acquired (**Fig. R6a**), we can calculate modulation depth by dividing the photothermal amplitude with concurrently acquire scattering intensity amplitude (particle intensity subtract background intensity). The corresponding modulation depth and estimated temperature rise is shown in **Fig.R6b**. For this D=500nm PMMA particle under mid-IR excitation at 1729 cm^{-1} , the measured highest temperature rise is 7.6 K.

Figure R6. Transient temperature rising of D=500 nm PMMA particle under mid-IR excitation at 1729 cm^{-1} . (a) Raw PDI signal from D=500 nm PMMA particle. Raw photothermal signal is separated into DC and AC components. AC is amplified 100 times. DC is directly digitized synchronically. (b) Modulation depth and calculated temperature rise from signal shown in (a). The modulation depth is calculated by dividing the AC signal amplitude with pure DC scattering intensity amplitude and amplification. Transient temperature is derived by dividing modulation depth with 0.39%/K, the average scattering intensity change percent per kelvin evaluated from Mie-scattering according to the method in reference¹⁴. The peak modulation depth is 3% according to the highest temperature rise of 7.6 K.

The estimation of temperature rise above is now added in **Supplementary S.3** of the supplementary document.

Response letter references

- 1 Bai, Y., Yin, J. & Cheng, J.-X. Bond-selective imaging by optically sensing the mid-infrared photothermal effect. *Science Advances* **7**, eabg1559 (2021).
- 2 Klementieva, O. *et al.* Super-Resolution Infrared Imaging of Polymorphic Amyloid Aggregates Directly in Neurons. *Advanced Science* **7**, 1903004 (2020).
- 3 Samolis, P. D. *et al.* Label-free imaging of fibroblast membrane interfaces and protein signatures with vibrational infrared photothermal and phase signals. *Biomedical optics express* **12**, 303-319 (2021).
- 4 Zhang, D. *et al.* Depth-resolved mid-infrared photothermal imaging of living cells and organisms with submicrometer spatial resolution. *Science advances* **2**, e1600521 (2016).
- 5 Kucsko, G. *et al.* Nanometre-scale thermometry in a living cell. *Nature* **500**, 54-58 (2013).
- 6 Mecklenburg, M. *et al.* Nanoscale temperature mapping in operating microelectronic devices. *Science* **347**, 629-632 (2015).
- 7 Chen, Z. *et al.* Imaging local heating and thermal diffusion of nanomaterials with plasmonic thermal microscopy. *ACS nano* **9**, 11574-11581 (2015).
- 8 Stringari, C. *et al.* Phasor approach to fluorescence lifetime microscopy distinguishes different metabolic states of germ cells in a live tissue. *Proceedings of the National Academy of Sciences* **108**, 13582-13587 (2011).
- 9 Datta, R., Heaster, T. M., Sharick, J. T., Gillette, A. A. & Skala, M. C. Fluorescence lifetime imaging microscopy: fundamentals and advances in instrumentation, analysis, and applications. *Journal of biomedical optics* **25**, 071203 (2020).
- 10 Dada, O. O., Feist, P. E. & Dovichi, N. J. Thermal diffusivity imaging with the thermal lens microscope. *Applied optics* **50**, 6336-6342 (2011).
- 11 Wang, L. & Xu, X. G. Scattering-type scanning near-field optical microscopy with reconstruction of vertical interaction. *Nature communications* **6**, 1-9 (2015).
- 12 Kim, B., Jahng, J., Sifat, A., Lee, E. S. & Potma, E. O. Monitoring Fast Thermal Dynamics at the Nanoscale through Frequency Domain Photoinduced Force Microscopy. *The Journal of Physical Chemistry C* **125**, 7276-7286 (2021).
- 13 Pavlovec, I. M. *et al.* Infrared photothermal heterodyne imaging: Contrast mechanism and detection limits. *Journal of Applied Physics* **127**, 165101 (2020).
- 14 Li, Z., Aleshire, K., Kuno, M. & Hartland, G. V. Super-resolution far-field infrared imaging by photothermal heterodyne imaging. *The Journal of Physical Chemistry B* **121**, 8838-8846 (2017).

REVIEWERS' COMMENTS

Reviewer #1 (Remarks to the Author):

The authors properly addressed my concerns and now I am convinced it is worth publishing in NC.

Reviewer #2 (Remarks to the Author):

I think the authors did a great job addressing the comments and thoroughly updated their manuscript.

As such, the revised version should be ready for publication.

Reviewer #3 (Remarks to the Author):

Yin et al. have revised their manuscript based on the reviewer comments. All of my comments have been addressed satisfactorily. They have extended the text and figures. Overall, the suggested technique is an incremental extension of the MIP microscopy. The new information that can be obtained by the technique is very valuable in various contexts, which is now better displayed by the manuscript.

I can therefore recommend the manuscript for publication in Nature Communications.

Reviewer #1 (Remarks to the Author):

The authors properly addressed my concerns and now I am convinced it is worth publishing in NC.

Re: We deeply appreciate the positive feedback from the reviewer.

Reviewer #2 (Remarks to the Author):

I think the authors did a great job addressing the comments and thoroughly updated their manuscript.

As such, the revised version should be ready for publication.

Re: We deeply appreciate the positive feedback from the reviewer.

Reviewer #3 (Remarks to the Author):

Yin et al. have revised their manuscript based on the reviewer comments. All of my comments have been addressed satisfactorily. They have extended the text and figures. Overall, the suggested technique is an incremental extension of the MIP microscopy. The new information that can be obtained by the technique is very valuable in various contexts, which is now better displayed by the manuscript.

I can therefore recommend the manuscript for publication in Nature Communications.

Re: We deeply appreciate the positive feedback from the reviewer.